# ⌨ ConvCodeWorld: Benchmarking Conversational Code Generation in Reproducible Feedback Environments

**Hojae Han**◇* **Seung-won Hwang**◇ **Rajhans Samdani**♠ **Yuxiong He**♠
♠Snowflake AI Research ◇Seoul National University

## Abstract

Large language models (LLMs) have proven invaluable for code generation, particularly in interactive settings. However, existing code generation benchmarks fail to capture the diverse feedback encountered in multi-turn interactions, limiting our ability to evaluate LLMs in these contexts. To address this gap, we present a set of novel benchmarks that explicitly model the quality of feedback provided to code generation LLMs. Our contributions are three-fold: **First**, we introduce ConvCodeWorld, a novel and reproducible environment for benchmarking interactive code generation. ConvCodeWorld simulates 9 distinct interactive code generation scenarios while systematically combining three types of feedback: (a) compilation feedback; (b) execution feedback with varying test coverage; (c) verbal feedback generated by GPT-4o with different levels of expertise. **Second**, we introduce ConvCodeBench, a fast, static version of benchmark that uses pre-generated feedback logs, eliminating the need for costly dynamic verbal feedback generation while maintaining strong Spearman's rank correlations (0.82 to 0.99) with ConvCodeWorld. **Third**, extensive evaluations of both closed-source and open-source LLMs including R1-Distill on ConvCodeWorld reveal key insights: (a) LLM performance varies significantly based on the feedback provided; (b) Weaker LLMs, with sufficient feedback, can outperform single-turn results of state-of-the-art LLMs without feedback; (c) Training on a specific feedback combination can limit an LLM's ability to utilize unseen combinations; (d) LLMs solve problems in fewer turns (high MRR) may not solve as many problems overall (high Recall), and vice versa. All implementations and benchmarks are publicly available at `https://huggingface.co/spaces/ConvCodeWorld/ConvCodeWorld`.

## 1 Introduction

Human-AI pair programming has become a promising approach to boost software development productivity, where large language models (LLMs) iteratively refine the code from developers' feedback. However, most existing benchmarks focus on single-turn scenarios, where LLMs are expected to generate executable code in one attempt Chen et al. (2021); Hendrycks et al. (2021); Austin et al. (2021); Li et al. (2022); Zhuo et al. (2024).

To address these gaps, we introduce ConvCodeWorld (§3; left panel in Figure 1), a novel environment for benchmarking interactive multi-turn code generation across diverse feedback combinations. ConvCodeWorld features nine scenarios by combining three feedback types: (a) compilation feedback, (b) execution feedback with partial and full test coverage, and (c) novice and expert level verbal human feedback. We simulate human feedback using GPT-4o (OpenAI, 2024) to generate verbal responses, ensuring reproducibility and cost-efficiency at only 1.5% of the cost of human annotation (Appendix C.2).

While replacing expensive human intervention with LLMs in ConvCodeWorld reduces costs, it can still be expensive due to computational overhead or API fees, and latency due to LLM response.

---

*Work done while visiting at Snowflake AI Research. Correspond to seungwonh@snu.ac.kr

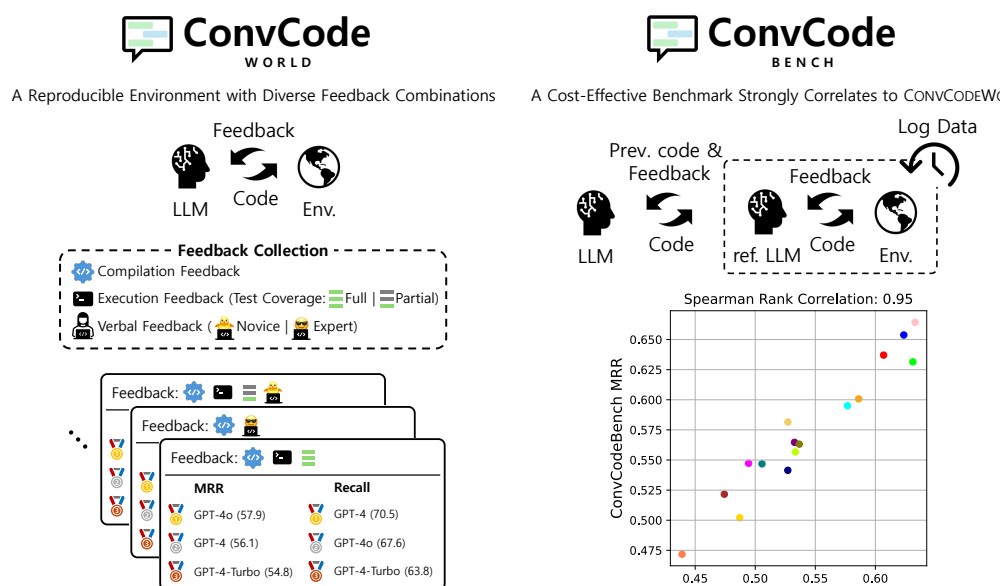

Figure 1: **(Left)** CONVCODEWORLD is a dynamic, reproducible environment that simulates nine distinct feedback scenarios by combining three types of feedback. **(Right)** CONVCODEBENCH is a static version of the benchmark that uses pre-generated logs and strongly correlates with CONVCODEWORLD. Together, these frameworks provide a comprehensive, cost-effective approach for evaluating LLMs in multi-turn, feedback-driven code generation, enabling scalable and consistent benchmarking across diverse feedback combinations.

To address these issues, we introduce CONVCODEBENCH (§4; right panel in Figure 1), a static benchmark using pre-generated feedback logs. CONVCODEBENCH eliminates the need for real-time feedback generation while maintaining strong correlation with CONVCODEWORLD (Spearman's rank 0.82-0.99; §5.3), offering a cost-effective and scalable solution for large-scale LLM benchmarking.

Existing benchmarks like InterCode (Yang et al., 2023) and MINT (Wang et al., 2024) lack the variety feedback combinations needed for comprehensive LLM performance assessment (§2). Additionally, their reliance on LLM calls for verbal feedback increases costs. Our study stands out by (a) offering a reproducible environment with **9 unique feedback combinations**, and (b) providing a **cost-effective benchmark** using pre-generated logs, avoiding costly LLM calls for verbal feedback while maintaining strong correlation with live results.

Through extensive experiments using both CONVCODEWORLD and CONVCODEBENCH across 21 different open and closed-source models including R1-Distill (DeepSeek-AI et al. (2025); Appendix A), we have gathered several key insights: (§5.2):

- **Feedback Combinations Diversifying Evaluation:** LLM performance varies across feedback settings, with feedback combinations affecting model rankings, highlighting the need for evaluation across diverse scenarios.

- **Weaker Models with Feedback Surpassing Single-Turn SOTA:** Weaker LLMs, with sufficient multi-turn feedback, can surpass state-of-the-art models in single-turn scenarios without feedback. This emphasizes the importance of interactive multi-turn code generation.

- **Generalization Challenges:** Models trained on limited feedback struggle to generalize to unseen combinations, highlighting the difficulty of adapting LLMs to new scenarios.

- **MRR and Recall Trade-off:** LLMs that efficiently solve problems in fewer turns (high MRR) may not solve as many problems in total (high Recall), highlighting a trade-off between efficiency and problem coverage.

## 2 RELATED WORK

Code generation benchmarks have traditionally focused on single-turn generation from natural language problem descriptions (Chen et al., 2021; Austin et al., 2021; Li et al., 2022; Zhuo et al., 2024). More recently, LLM performance has improved through interactions with external tools, such as interpreters for compiling, executing test cases, and verbal feedback, resulting in more accurate outputs (Shinn et al., 2023; Madaan et al., 2024; Chen et al., 2024; Olausson et al., 2024). This shift has led to the development of multi-turn benchmarks like InterCode (Yang et al., 2023) and MINT (Wang et al., 2024).

However, existing multi-turn benchmarks remain limited in feedback diversity. InterCode focuses on compilation and partial execution feedback but lacks full test coverage and verbal feedback. MINT generates verbal feedback via GPT-4, reducing human-in-the-loop evaluation costs, but its feedback scope is narrow and requires costly LLM calls for each evaluation.

Our study presents (a) CONVCODEWORLD, a reproducible environment with **nine unique feedback combinations** (Table 1), and (b) CONVCODEBENCH, a **cost-effective benchmark** that maintains high correlation with live environment by using pre-generated logs, elim-

Table 1: Feedback combinations ($\Omega$; §3.2) across InterCode, MINT and CONVCODEWORLD, constructed by different feedback types (§3.1).

| $\Omega$ | InterCode | MINT | CONVCODEWORLD |
|---|---|---|---|
| $\langle f_c, \phi, \phi \rangle$ | ✗ | ✗ | ✓ |
| $\langle f_c, f_e, \phi \rangle$ | ✗ | ✓ | ✓ |
| $\langle f_c, f_e^*, \phi \rangle$ | ✓ | ✗ | ✓ |
| $\langle f_c, \phi, f_v \rangle$ | ✗ | ✗ | ✓ |
| $\langle f_c, f_e, f_v \rangle$ | ✗ | ✓ | ✓ |
| $\langle f_c, f_e^*, f_v \rangle$ | ✗ | ✗ | ✓ |
| $\langle f_c, \phi, f_v^* \rangle$ | ✗ | ✗ | ✓ |
| $\langle f_c, f_e, f_v^* \rangle$ | ✗ | ✓ | ✓ |
| $\langle f_c, f_e^*, f_v^* \rangle$ | ✗ | ✗ | ✓ |

inating the need for costly LLM calls to provide verbal feedback. We further discuss the distinction of CONVCODEWORLD in Appendix B.

## 3 CONVCODEWORLD: REPRODUCIBLE FEEDBACK ENVIRONMENTS

In real-world settings of interactive code generation, the types and combinations of feedback can vary significantly due to factors such as the availability of feedback from code execution (e.g., error messages, output) and the expertise of the feedback provider. These variations, particularly the provider's expertise, can strongly influence the quality of the verbal feedback when it is offered.

To effectively evaluate LLMs under these varying conditions, we propose CONVCODEWORLD, a novel and reproducible environment designed to simulate a wide range of interactive code generation scenarios. Two key features of CONVCODEWORLD are as follows: (a) **Encompassing Diverse Real-World Scenarios:** CONVCODEWORLD covers nine distinct feedback combinations that occur in practical settings; (b) **Ensure the Reproducibility of Evaluation:** CONVCODEWORLD provides a consistent and repeatable framework for assessing the performance of LLMs.

Table 2: By providing diverse feedback types, with different coverage levels in execution and natural language feedback, ours encompasses a broader range of realistic scenarios. △ indicates partial coverage (§3.1) with specific limitations: [1]Syntax errors only, [2]Limited by test coverage or feedback provider, and [3]Potential misguidance due to limited expertise.

| Feedback | Fault Localization | Guidance for Refinement |
|---|---|---|
| $f_c$ | △[1] | ✗ |
| $f_e$ | △[2] | ✗ |
| $f_e^*$ | ✓ | ✗ |
| $f_v$ | △[2] | △[3] |
| $f_v^*$ | ✓ | ✓ |

### 3.1 FEEDBACK CATEGORIZATION

To accurately simulate real-world feedback in interactive code generation, we focus on two critical components: (a) **Fault Localization:** Identifying the specific parts of the code where issues or errors occur; (b) **Guidance for Refinement:** Offering suggestions or instructions on how to correct the identified issues.

As means of obtaining such information, we consider three different types of feedback: compilation feedback, execution feedback, and verbal feedback.

**Compilation Feedback ($f_c$)** Originated from the compiler, this feedback identifies syntax and type-checking errors but cannot localize logical or runtime errors. As a result, Table 2 marks this with $\triangle$ for partial fault localization. Additionally, compilation errors do not offer refinement guidance.

**Execution Feedback** Derived from code execution, this feedback includes runtime errors and test run results. Full or partial fault localization is provided, depending on test coverage (TC): (a) **Full TC ($f_e^*$):** Inspired by Test-Driven Development (TDD; Beck, 2022), complete test cases allow precise fault localization, identifying where and under what conditions the code fails. This provides details on the failure's location and triggering inputs; (b) **Partial TC ($f_e$):** In more realistic settings with partial test coverage, fault localization is limited to tested code lines, potentially leaving faults in untested sections undetected. This type of feedback simulates incomplete real-world test suites, where only a subset of possible execution paths is covered. Refinement guidance is not provided in either full or partial test coverage executions.

**Verbal Feedback** Verbal feedback in our benchmark is generated by LLMs simulating human feedback, ranging from novice to expert levels. This feedback could emulate responses from humans, such as experts guiding LLMs to generate code, or novices without coding expertise. Both fault localization and refinement guidance are provided verbally, but the extent and accuracy of this feedback depend on the simulated provider: (a) **Novice-Level ($f_v$):** At this level, the LLM simulates novice feedback, which tends to rely heavily on other feedback types (e.g., compilation or execution feedback) and often restates observed errors without deeper understanding. Refinement guidance may be incorrect or absent, due to the novice's limited expertise simulated by the LLM's potential hallucinations. (b) **Expert-Level ($f_v^*$):** Expert feedback reflects scenarios where expert programmers use LLMs to automate simpler tasks, allowing them to concentrate on more complex coding challenges. This feedback is simulated by the LLM to provide detailed fault localization and code refinement guidance. It generates the feedback an expert programmer might give, focusing on resolving issues with a deep understanding of programming concepts and the expected functionality.

### 3.1.1 VERBAL FEEDBACK GENERATION

We generate $f_v$ and $f_v^*$ by GPT-4o with in-context learning (Dong et al., 2022). We chose GPT-4o as we found it to be best at following instructions and minimizing risks such as ground truth code leakage, as discussed in Appendix C.5.

- **Generation of $f_v$:** Novice-level verbal feedback is constructed by verbalizing outputs from compilation and/or execution feedback, possibly supplemented with language model predictions.

- **Generation of $f_v^*$:** Expert-level verbal feedback is produced by showing the agent's code with the correct reference code (Wang et al., 2024), enabling a comparison and subsequent detailed feedback on required modifications. We perform extensive analysis to ensure no ground truth code is leaked during $f_v^*$ generation (see Appendix C.5 for analysis on this).

See Appendices C.6 for comparative analysis of verbal feedback using different LLMs, I for the in-context examples, and J for a generated example of $f_v^*$. For the detailed prompting methods for the $f_v$ and $f_v^*$ construction, please refer to our codebase.[1]

**Reproducibility and Cost-Efficiency Compared to Human Annotation** Manual annotation of verbal feedback is costly and lacks reproducibility. Instead, we use GPT-4o, as supported by prior studies demonstrating the effectiveness of LLM-generated feedback in benchmarks (Wang et al., 2024; Yao et al., 2024). This approach improves reproducibility by using a consistent feedback provider and reduces annotation costs to about 1.5% (Appendix C.2) of those for human annotators.

### 3.2 FEEDBACK COMBINATIONS

In each of our turns, we simulate different real-world interactive code generation scenarios by combining representative feedback combinations. We represent feedback settings by taking a Cartesian product across compilation feedback settings, execution feedback settings, and verbal feedback settings. In particular, we formalize a feedback combination $\Omega$ as a tuple of feedback expressed by

---

[1] `https://huggingface.co/spaces/ConvCodeWorld/ConvCodeWorld`

regular expression notation:

$$\Omega = \langle f_c, [\phi|f_e|f_e^*], [\phi|f_v|f_v^*] \rangle. \tag{1}$$

The choices of feedback settings is simply dictated by these observations: (a) Compilation feedback $f_c$ is always present since it is cheap and universally available; (b) Execution feedback varies among being unavailable ($\phi$), available with partial test coverage ($f_e$), or with full test coverage ($f_e^*$); (c) Verbal feedback can be also unavailable ($\phi$), available with novice-level ($f_v$), or with expert-level ($f_v^*$). By combining these options—1 for compilation feedback, 3 for execution feedback, and 3 for verbal feedback—we obtain 9 distinct feedback combinations.

Each feedback combination $\Omega$ reflects a unique real-world scenario, allowing us to comprehensively evaluate LLMs under diverse conditions as listed in Table 1.

Now it is easy to formalize the interactive code generation in CONVCODEWORLD: For each turn $t$, the target code generation model $\mathcal{M}$ iteratively generates the next version of code $\mathcal{C}_{t+1}^{\mathcal{M}}$ from the problem description $x$, the generated code $\mathcal{C}_t^{\mathcal{M}}$, and the corresponding tuple of feedback $\Omega_t$:

$$\mathcal{C}_{t+1}^{\mathcal{M}} = \mathcal{M}(x; \mathcal{C}_t^{\mathcal{M}}; \Omega_t). \tag{2}$$

## 4 CONVCODEBENCH: A STATIC BENCHMARK FOR EFFICIENT EVALUATION

While CONVCODEWORLD provides a comprehensive live benchmark for evaluating LLMs in interactive code generation scenarios, it requires access to an LLM for verbal feedback generation. Although this approach is more efficient and reproducible than using human annotators, it still introduces additional overhead, cost, and potential reproducibility issues, especially when using closed API models like GPT-4o. To address these challenges, we propose CONVCODEBENCH, a static benchmark designed to complement CONVCODEWORLD.

CONVCODEBENCH leverages feedback logs generated by a fixed reference model interacting with GPT-4o. The benchmark presents pre-generated conversations—including the code produced by the reference model and the corresponding feedback, such as verbal feedback by GPT-4o—and tasks the target code model with refining the code. We revise Equation 2 to formalize CONVCODEBENCH as follows. For each turn $t$, the target code generation model $\mathcal{M}$ is provided generated code $\mathcal{C}_t^{\overline{\mathcal{M}}}$ from a reference model $\overline{\mathcal{M}}$, and the corresponding tuple of feedback $\bar{\Omega}_t$ provided to outputs generated by $\overline{\mathcal{M}}$. Given the model and feedback corresponding to a reference model, the target model $\mathcal{M}$ generates the next version of code $\mathcal{C}_{t+1}^{\mathcal{M}}$:

$$\mathcal{C}_{t+1}^{\mathcal{M}} = \mathcal{M}(x; \mathcal{C}_t^{\overline{\mathcal{M}}}; \bar{\Omega}_t). \tag{3}$$

This approach offers several advantages:

- **Elimination of Dependency on External LLMs or APIs for Verbal Feedback Generation:** By using static feedback logs, CONVCODEBENCH reduces costs and latency associated with real-time LLM interactions.
- **Parallel Processing of Inference Calls:** The static nature of the benchmark allows for batched evaluation requests across all turns, enabling faster turnaround times.
- **Enhanced Reproducibility:** Utilizing fixed logs ensures consistent evaluations, further increasing reproducibility.

One key concern when using CONVCODEBENCH is the bias introduced by pre-generated interaction logs prompting the question: *Can we ensure high correlation between static and live benchmarks by an appropriate choice of reference model?*

We hypothesize that using logs from a weaker model, where the generated code still requires refinement even after multiple turns, allows for better differentiation among models based on their ability to improve unsolved code.

Table 3: Performance of three different LLMs at turn 0 (i.e. the initial code generation without feedback) and at turn 10 on CONVCODEWORLD where $\Omega = \langle f_c, \phi, f_v^* \rangle$.

| Model | Pass@1 | |
|---|---|---|
| | Turn 0 | Turn 10 |
| CodeLlama-7B-Instruct | 21.8 | 55.2 |
| DeepSeek-Coder-6.7B-Instruct | 35.2 | 83.1 |
| GPT-4-0613 | 46.0 | 92.5 |

Based on this rationale, we used CodeLlama-7B-Instruct as a reference model, as it is worse than many other models at both turns 0 and 10 (see Table 3). We find that creating CONVCODEBENCH with this model yields a very strong correlations with live settings. When comparing models on CONVCODEWORLD and CONVCODEBENCH, we obtained Spearman's rank correlations between 0.82 and 0.99. We find that using CodeLlama-7B-Instruct as the base model outperforms both DeepSeek-Coder-6.7B-Instruct (a stronger code model) and GPT-4 (one of the state-of-the-arts) as reference models (§5.3).

In summary, we find that CONVCODEBENCH is a great way of comparing code models within the framework of CONVCODEWORLD despite relying on logs from a reference model because of strong rank correlations across the two setups.

## 5 EXPERIMENTS

Using CONVCODEWORLD and CONVCODEBENCH, we conduct comprehensive experiments to evaluate LLMs' interactive code generation capabilities across diverse feedback combinations. This section outlines our experimental setup (§5.1), results on CONVCODEWORLD (§5.2), and results on CONVCODEBENCH (§5.3).

### 5.1 SETUP

To implement CONVCODEWORLD, we extended BigCodeBench-Full-Instruct (Zhuo et al., 2024), a single-turn Python code generation benchmark, into an interactive framework using a custom prompt pipeline built using DSPy (Khattab et al., 2024) (see Appendix E for the implementation details). BigCodeBench was selected for three key reasons: (a) its highly challenging problem sets (as of the writing of this paper, the highest performance on this data is 51.1% of Pass@1); (b) its large scale, with 1,140 problems, offering higher generalizability than smaller benchmarks like HumanEval (Chen et al., 2021; 164 problems) and MBPP-sanitized (Austin et al., 2021; 399-427 problems); and (c) its comprehensive test coverage—an average of 5.6 cases per problem with 99% branch coverage—enabling the evaluation of a wide spectrum of execution feedback scenarios, ranging from partial to full test coverage.

**Evaluation Metrics** In the interactive scenario, where code is iteratively refined based on feedback, we focus on two aspects for evaluation: (a) the number of turns it takes to produce correct code, with fewer turns being preferable, and (b) whether the model can eventually solve the problem within a set number of turns $n$. In our experiments, we set $n = 10$.

To capture these aspects, we use Pass@1 (Chen et al., 2021) as the core metric to assess code correctness at each turn and adapt two complementary metrics from information retrieval: (a) **Mean Reciprocal Rank (MRR):** $\frac{1}{k}$ where $k$ is the turn at which the model produces correct code. If no correct code is generated within $n$ turns, the score is set to 0; (b) **Recall:** 1 if the model produces correct code within $n$ turns.

**Baseline LLMs** We extensively evaluated 3 closed-source and 18 open-source LLMs ranging from 7B to 70B:[2] (a) **Closed-Source:** We select three OpenAI LLMs—GPT-4-0613, GPT-4-Turbo-2024-04-09, and GPT-4o-2024-05-13; (b) **Open-Source:** Llama-3.1-70B-Instruct (Dubey et al., 2024), Llama-3.1-8B-Instruct, DeepSeek-Coder-V2-Lite-Instruct (Zhu et al. (2024); an MoE model; total params: 16B; active params: 2.4B), DeepSeek-Coder-33B-Instruct (Guo et al., 2024), DeepSeek-Coder-6.7B-Instruct, ReflectionCoder-DS-33B (Ren et al., 2024), ReflectionCoder-DS-6.7B, Qwen1.5-72B-Chat (Bai et al., 2023), Qwen1.5-32B-Chat, CodeQwen1.5-7B-Chat, StarCoder2-15B-Instruct-v0.1,[3] CodeLlama-34B-Instruct (Roziere et al., 2023),[4] CodeLlama-13B-Instruct, and CodeLlama-7B-Instruct. In Appendix A, we further included the results of two recent R1-Distill (DeepSeek-AI et al., 2025) models and their base models on CONVCODEWORLD.

---

[2]While we attempted smaller models like DeepSeek-Coder-1.3B-Instruct, it failed to follow interactive code generation format, resulting degeneration.

[3]https://huggingface.co/bigcode/starcoder2-15b-instruct-v0.1

[4]We excluded CodeLlama-70B-Instruct due to its 4K token length limitation, which is too short for interactive code generation.

Table 4: MRR results on CONVCODEWORLD. ✗ indicates that no feedback of that type is provided ($\phi$). The leftmost results, with three ✗, represent $\Omega = \langle \phi, \phi, \phi \rangle$, corresponding to single-turn code generation without any feedback. For each column, bold and underscore indicate 1st and 2nd place performance within the same model group.

| Compilation Feedback | ✗ | $f_c$ | $f_c$ | $f_c$ | $f_c$ | $f_c$ | $f_c$ | $f_c$ | $f_c$ | $f_c$ |
|---|---|---|---|---|---|---|---|---|---|---|
| Execution Feedback | ✗ | ✗ | $f_e$ | $f_e^*$ | ✗ | $f_e$ | $f_e^*$ | ✗ | $f_e$ | $f_e^*$ |
| Verbal Feedback | ✗ | ✗ | ✗ | ✗ | $f_v$ | $f_v$ | $f_v$ | $f_v^*$ | $f_v^*$ | $f_v^*$ |
| *Closed-Source Models* | | | | | | | | | | |
| GPT-4-0613 | 46.0 | 46.0 | 52.1 | 56.1 | 46.0 | 52.4 | 56.4 | 63.1 | 64.3 | 64.8 |
| GPT-4-Turbo-2024-04-09 | 48.0 | 48.0 | 51.8 | 54.8 | 48.0 | 52.6 | 56.4 | 62.4 | 64.3 | 64.5 |
| GPT-4o-2024-05-13 | 50.8 | 50.8 | 55.0 | 57.9 | 50.8 | 55.1 | 58.6 | 63.3 | 64.7 | 65.3 |
| *Open-Source Models (⩾ 30B)* | | | | | | | | | | |
| Llama-3.1-70B-Instruct | 45.4 | 45.4 | 49.9 | 53.4 | 45.4 | 50.8 | 55.2 | 60.7 | 62.6 | 63.3 |
| DeepSeek-Coder-33B-Instruct | 41.6 | 41.6 | 43.4 | 43.6 | 41.6 | 45.5 | 48.0 | 58.6 | 58.5 | 58.8 |
| ReflectionCoder-DS-33B | 41.6 | 41.6 | 42.9 | 42.9 | 41.6 | 45.6 | 48.1 | 57.7 | 58.2 | 58.9 |
| Qwen1.5-72B-Chat | 32.9 | 33.0 | 35.8 | 38.3 | 33.0 | 38.6 | 41.4 | 50.6 | 52.0 | 52.7 |
| Qwen1.5-32B-Chat | 32.0 | 32.0 | 35.3 | 36.7 | 32.0 | 36.6 | 39.7 | 47.4 | 42.6 | 40.8 |
| CodeLlama-34B-Instruct | 28.8 | 28.8 | 31.0 | 31.9 | 28.8 | 32.5 | 35.1 | 48.7 | 49.2 | 49.8 |
| *Open-Source Models (< 30B)* | | | | | | | | | | |
| Llama-3.1-8B-Instruct | 31.4 | 31.5 | 34.0 | 34.6 | 31.5 | 36.1 | 39.1 | 49.4 | 49.8 | 51.3 |
| DeepSeek-Coder-V2-Lite-Instruct | 38.3 | 38.3 | 40.5 | 41.7 | 38.3 | 42.0 | 43.8 | 52.7 | 52.9 | 53.3 |
| DeepSeek-Coder-6.7B-Instruct | 35.2 | 35.2 | 36.2 | 36.1 | 35.2 | 38.8 | 40.5 | 53.3 | 53.2 | 53.9 |
| ReflectionCoder-DS-6.7B | 37.4 | 37.4 | 38.3 | 38.7 | 37.4 | 40.4 | 42.4 | 53.3 | 53.8 | 53.6 |
| CodeQwen1.5-7B-Chat | 39.3 | 39.4 | 39.7 | 40.1 | 39.3 | 42.0 | 43.7 | 53.7 | 53.5 | 54.8 |
| StarCoder2-15B-Instruct-v0.1 | 37.1 | 37.1 | 37.9 | 38.3 | 37.1 | 39.4 | 40.5 | 52.7 | 52.8 | 52.1 |
| CodeLlama-13B-Instruct | 28.4 | 28.4 | 29.0 | 29.0 | 28.4 | 31.2 | 33.0 | 43.9 | 44.3 | 44.8 |
| CodeLlama-7B-Instruct | 21.8 | 21.8 | 22.3 | 22.3 | 21.8 | 23.5 | 25.2 | 35.0 | 33.4 | 33.9 |

## 5.2 RESULTS ON CONVCODEWORLD

Tables 4 and 5 present MRR and Recall scores, respectively, for both closed-source and open-source LLMs across various feedback combinations. These results provide a comprehensive view of model performance in CONVCODEWORLD.

**Overview of Results**    While closed-source models generally outperformed most open-source models, Llama-3.1-70B-Instruct demonstrated competitive Recall performance, surpassing both GPT-4-Turbo and GPT-4o in certain scenarios like $\langle f_c, [f_e | f_e^*], f_v \rangle$ and $\langle f_c, [\phi | f_e | f_e^*], f_v^* \rangle$.

Notably, this Recall gap between closed-source and open-source models narrows significantly under specific feedback settings, particularly when expert-level verbal feedback $f_v^*$ is provided. For instance, in the $\langle f_c, \phi, f_v^* \rangle$ setting, DeepSeek-Coder6.7B-Instruct (82.8) outperformed GPT-4o (82.3), and DeepSeek-Coder33B-Instruct (85.4) outperformed GPT-4-Turbo (84.7).

Another key observation is that, among open-source models smaller than 30B, no clear winner emerges across all feedback combinations. This emphasizes the importance of selecting models based on the specific type of feedback available.

### 5.2.1 FEEDBACK COMBINATIONS: DIVERSIFIED EVALUATION

We observed significant performance variation within the same model across different feedback combinations, emphasizing the necessity of CONVCODEWORLD for evaluating code generation models under diverse feedback conditions.

Specifically, we summarize the effect of providing different feedback combinations:

**Impact of Novice-Level Verbal Feedback on Execution Feedback Utilization**    Without novice-level verbal feedback ($f_v$), some models—DeepSeek-Coder-33B-Instruct, DeepSeek-Coder-6.7B-Instruct, CodeQwen1.5-7B-Chat, StarCoder2-15B-Instruct-v0.1, CodeLlama-13B-Instruct, and CodeLlama-7B-Instruct—showed minimal performance differences between partial ($\langle f_c, f_e, \phi \rangle$) and full ($\langle f_c, f_e^*, \phi \rangle$) test coverage in execution feedback. However, these models showed greater reliance on $f_v$, especially in $\langle f_c, f_e^*, f_v \rangle$ compared to $\langle f_c, f_e, f_v \rangle$, indicating that they need $f_v$ to fully leverage $f_e^*$. In contrast, high-performing models—GPT-4, GPT-4-Turbo, GPT-4o, and Llama-3.1-70B—demonstrated a larger performance boost from $\langle f_c, f_e, \phi \rangle$ to $\langle f_c, f_e^*, \phi \rangle$ compared to the boost

Table 5: Recall results on CONVCODEWORLD. ✗ indicates that no feedback of that type is provided ($\phi$). The leftmost results, with three ✗, represent $\Omega = \langle \phi, \phi, \phi \rangle$, corresponding to single-turn code generation without any feedback. For each column, bold and underscore indicate 1st and 2nd place performance within the same model group.

| Compilation Feedback | ✗ | $f_c$ | $f_c$ | $f_c$ | $f_c$ | $f_c$ | $f_c$ | $f_c$ | $f_c$ | $f_c$ |
|---|---|---|---|---|---|---|---|---|---|---|
| Execution Feedback | ✗ | ✗ | $f_e$ | $f_e^*$ | ✗ | $f_e$ | $f_e^*$ | ✗ | $f_e$ | $f_e^*$ |
| Verbal Feedback | ✗ | ✗ | ✗ | ✗ | $f_v$ | $f_v$ | $f_v$ | $f_v^*$ | $f_v^*$ | $f_v^*$ |
| Closed-Source Models | | | | | | | | | | |
| GPT-4-0613 | 46.0 | 46.0 | 60.3 | 70.5 | 46.0 | 61.9 | 72.5 | 89.7 | 91.1 | 92.5 |
| GPT-4-Turbo-2024-04-09 | 48.0 | 48.0 | 56.7 | 63.8 | 48.0 | 58.6 | 68.1 | 84.7 | 87.5 | 88.5 |
| GPT-4o-2024-05-13 | 50.8 | 50.8 | 60.5 | 67.6 | 50.8 | 60.8 | 69.6 | 82.3 | 84.9 | 86.2 |
| Open-Source Models ($\geqslant$ 30B) | | | | | | | | | | |
| Llama-3.1-70B-Instruct | 45.4 | 45.4 | 56.2 | 64.8 | 45.4 | 59.5 | 70.8 | 86.7 | 88.9 | 91.8 |
| DeepSeek-Coder-33B-Instruct | 41.6 | 41.6 | 45.5 | 46.1 | 41.6 | 50.4 | 56.6 | 85.4 | 84.6 | 85.6 |
| ReflectionCoder-DS-33B | 41.6 | 41.6 | 45.3 | 44.9 | 41.6 | 51.4 | 57.2 | 81.4 | 81.8 | 84.2 |
| Qwen1.5-72B-Chat | 32.9 | 33.2 | 39.9 | 47.5 | 33.2 | 47.5 | 57.9 | 84.4 | 86.1 | 87.2 |
| Qwen1.5-32B-Chat | 32.0 | 32.0 | 41.1 | 45.3 | 32.0 | 44.6 | 54.3 | 75.9 | 61.8 | 57.1 |
| CodeLlama-34B-Instruct | 28.8 | 28.8 | 33.7 | 35.8 | 28.8 | 37.5 | 44.6 | 80.0 | 82.0 | 82.3 |
| Open-Source Models (< 30B) | | | | | | | | | | |
| Llama-3.1-8B-Instruct | 31.4 | 31.8 | 38.4 | 40.0 | 31.7 | 43.2 | 51.8 | 80.9 | 80.2 | 83.7 |
| DeepSeek-Coder-V2-Lite-Instruct | 38.3 | 38.3 | 43.4 | 46.1 | 38.3 | 47.0 | 51.4 | 76.3 | 75.8 | 76.9 |
| DeepSeek-Coder-6.7B-Instruct | 35.2 | 35.2 | 37.7 | 37.5 | 35.2 | 43.3 | 48.2 | 82.8 | 82.5 | 83.1 |
| ReflectionCoder-DS-6.7B | 37.4 | 37.4 | 39.6 | 40.7 | 37.4 | 44.7 | 50.4 | 79.1 | 79.6 | 78.9 |
| CodeQwen1.5-7B-Chat | 39.3 | 39.6 | 40.1 | 41.1 | 39.5 | 45.8 | 49.5 | 74.4 | 74.7 | 77.4 |
| StarCoder2-15B-Instruct-v0.1 | 37.1 | 37.1 | 39.3 | 40.0 | 37.1 | 42.6 | 46.3 | 76.9 | 76.8 | 75.6 |
| CodeLlama-13B-Instruct | 28.4 | 28.4 | 29.7 | 30.0 | 28.4 | 35.1 | 41.1 | 69.0 | 70.7 | 71.6 |
| CodeLlama-7B-Instruct | 21.8 | 21.8 | 22.9 | 23.0 | 21.8 | 26.2 | 30.5 | 61.7 | 53.9 | 55.2 |

from $\langle f_c, f_e, \phi \rangle$ to $\langle f_c, f_e, f_v \rangle$. This suggests these models can infer refinement strategies directly from raw execution feedback without heavily relying on $f_v$.

**Impact of Expert-Level Verbal Feedback on Execution Feedback Utilization** Most models demonstrated performance improvements with richer execution feedback, progressing through the sequences $\langle f_c, \phi, f_v^* \rangle$, $\langle f_c, f_e, f_v^* \rangle$, and $\langle f_c, f_e^*, f_v^* \rangle$. However, exceptions arise: (a) DeepSeek-Coder family and ReflectionCoder-DS-6.7B exhibited no performance difference with the inclusion of execution feedback; (b) Llama-3.1-8B-Instruct, ReflectionCoder-DS-33B, and CodeQwen1.5-7B-Chat showed no significant difference between $\langle f_c, \phi, f_v^* \rangle$ and $\langle f_c, f_e, f_v^* \rangle$, but performance improved when full test coverage ($\langle f_c, f_e^*, f_v^* \rangle$) was ensured; (c) In some weaker models—Qwen1.5-32B-Chat and StarCoder2-15B-Instruct-v0.1—increasing the test coverage from $\langle f_c, f_e, f_v^* \rangle$ to $\langle f_c, f_e^*, f_v^* \rangle$ resulted in negative performance impacts. Additionally, the highest performance of Qwen1.5-32B-Chat was observed with $\langle f_c, \phi, f_v^* \rangle$, while adding execution feedback ($f_e$ or $f_e^*$) led to decreased performance. We hypothesize that weaker models struggle to utilize complex feedback effectively, resulting in lower performance. We further discuss the possible reasons for these exceptions in Appendix C.4.

### 5.2.2 MULTI-TURN FEEDBACK: WEAKER MODELS OUTPERFORMING SINGLE-TURN SOTA

Weaker LLMs with sufficient feedback outperformed the single-turn, no-feedback performance ($\langle \phi, \phi, \phi \rangle$) of state-of-the-art models like GPT-4 and GPT-4-Turbo.

**MRR** When expert-level verbal feedback ($f_v^*$) was incorporated, most weaker models, including DeepSeek-Coder-6.7B-Instruct and Llama-3.1-8B-Instruct, surpassed the single-turn code generation performance of state-of-the-art single-turn models such as GPT-4, GPT-4-Turbo, and GPT-4o. Additionally, with the inclusion of novice-level verbal feedback ($f_v$) and either partial or full execution feedback ($f_e$ or $f_e^*$), DeepSeek-Coder-33B-Instruct and ReflectionCoder-DS-33B matched or exceeded the single-turn performance of GPT-4 and GPT-4-Turbo.

**Recall** Most open-source models exhibited significant improvements when novice-level verbal feedback with execution feedback ($\langle f_c, [f_e | f_e^*], f_v \rangle$) or expert-level verbal feedback ($\langle f_c, [\phi | f_e | f_e^*], f_v \rangle$) was provided. Remarkably, providing execution feedback with full test coverage while omitting any verbal feedback ($\langle f_c, f_e^*, \phi \rangle$) enabled some models, such as DeepSeek-Coder-V2-Lite-Instruct, DeepSeek-Coder-33B-Instruct, and Qwen1.5-72B-Chat, to achieve or even exceed GPT-4's single-turn performance.

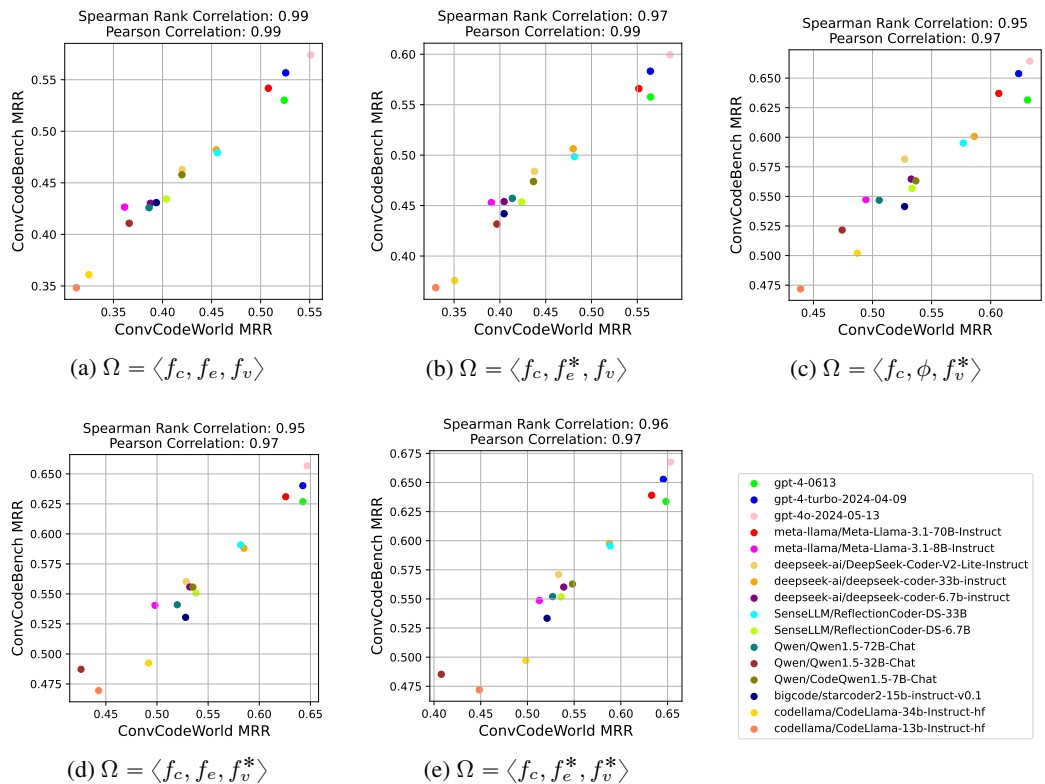

Figure 2: Correlation between MRR on CONVCODEBENCH (ref. CodeLlama-7B-Instruct) and MRR on CONVCODEWORLD with different feedback combinations $\Omega$.

### 5.2.3 GENERALIZATION: UNSEEN FEEDBACK COMBINATION

ReflectionCoder-DS family were initialized from DeepSeek-Coder-Instruct, and trained to refine code on a specific scenario of $\langle f_c, f_e^*, f_v \rangle$. As a result, ReflectionCoder-DS-6.7B outperformed DeepSeek-Coder-6.7B-Instruct on $\langle f_c, [f_e|f_e^*], f_v \rangle$. However, with unseen feedback like expert-level verbal feedback ($f_v^*$), the performance gap narrows significantly, with minimal MRR difference and DeepSeek-Coder-Instruct generally outperforming in Recall. This tendency is more pronounced in ReflectionCoder-DS-33B; except for $\langle f_c, [f_e|f_e^*], f_v \rangle$, ReflectionCoder-DS-33B consistently performed at or below the level of DeepSeek-Coder-33B-Instruct across all feedback combinations in both MRR and Recall. This indicates that training on a specific feedback combination can reduce the performance on the other combinations.

### 5.2.4 TRADE-OFF: MULTI-TURN MRR AND RECALL

We observed that an LLM requiring fewer turns to solve a problem (high MRR) may not excel at solving as many problems as possible (high Recall), and vice versa: (a) **Closed-Source Models:** GPT-4o achieved the highest MRR, while GPT-4 had the best Recall;[5] (b) **Open-Source Models ⩾ 30B:** Llama-3.1-70B led in both MRR and Recall. DeepSeek-Coder-33B-Instruct and ReflectionCoder-DS-33B followed in MRR. However, with $f_e^*$ or $f_v^*$ feedback, Qwen1.5-72B-Chat generally outperformed them in Recall, despite having a lower MRR; (c) **Open-Source Models < 30B:** MRR and Recall tendencies were similar without verbal feedback. With verbal feedback, CodeQwen1.5-7B-Chat excelled in MRR, while DeepSeek-Coder-V2-Lite-Instruct ($\langle f_c, [f_e|f_e^*], f_v \rangle$), and DeepSeek-Coder-6.7B-Instruct ($\langle f_c, [\phi|f_e|f_e^*], f_v^* \rangle$) led in Recall.

---

[5]This quantitatively confirms what some accounts observed on x.com

Table 6: MRR and Recall results on CONVCODEBENCH using logs of CodeLlama-7B-Instruct in CONVCODEWORLD. ✗ indicates that no feedback of that type is provided ($\phi$). For each column, bold and underscore indicate 1st and 2nd place performance within the same model group.

| | MRR | | | | | Recall | | | | |
|---|---|---|---|---|---|---|---|---|---|---|
| Compilation Feedback | $f_c$ | $f_c$ | $f_c$ | $f_c$ | $f_c$ | $f_c$ | $f_c$ | $f_c$ | $f_c$ | $f_c$ |
| Execution Feedback | $f_e$ | $f_e^*$ | ✗ | $f_e$ | $f_e^*$ | $f_e$ | $f_e^*$ | ✗ | $f_e$ | $f_e^*$ |
| Verbal Feedback | $f_v$ | $f_v$ | $f_v^*$ | $f_v^*$ | $f_v^*$ | $f_v$ | $f_v$ | $f_v^*$ | $f_v^*$ | $f_v^*$ |
| Closed-Source Models | | | | | | | | | | |
| GPT-4-0613 | 53.0 | 55.8 | 63.1 | 62.7 | 63.4 | 59.5 | 65.7 | 85.9 | **82.3** | 83.1 |
| GPT-4-Turbo-2024-04-09 | 55.7 | 58.3 | 65.4 | 64.0 | 65.3 | 61.8 | **68.2** | **86.8** | 81.4 | 84.2 |
| GPT-4o-2024-05-13 | **57.4** | **59.9** | **66.4** | **65.7** | **66.8** | **62.1** | 68.1 | 86.2 | 81.9 | **84.7** |
| Open-Source Models ($\geqslant$ 30B) | | | | | | | | | | |
| Llama-3.1-70B-Instruct | **54.2** | **56.6** | **63.7** | **63.1** | **63.9** | **60.2** | **65.7** | **85.9** | **81.5** | **84.0** |
| DeepSeek-Coder-33B-Instruct | 48.2 | 50.6 | 60.1 | 58.8 | 59.8 | 51.9 | 58.0 | 83.2 | 78.2 | 79.7 |
| ReflectionCoder-DS-33B | 47.9 | 49.9 | 59.5 | 59.1 | 59.6 | 51.2 | 56.2 | 82.2 | 77.8 | 79.6 |
| Qwen1.5-72B-Chat | 42.6 | 45.7 | 54.7 | 54.1 | 55.2 | 47.8 | 55.7 | 80.3 | 76.8 | 78.7 |
| Qwen1.5-32B-Chat | 41.1 | 43.2 | 52.2 | 48.7 | 48.5 | 45.7 | 51.4 | 76.2 | 67.2 | 66.8 |
| CodeLlama-34B-Instruct | 36.1 | 37.6 | 50.2 | 49.2 | 49.7 | 40.2 | 43.9 | 78.3 | 72.4 | 73.8 |
| Open-Source Models (< 30B) | | | | | | | | | | |
| Llama-3.1-8B-Instruct | 42.6 | 45.3 | 54.7 | 54.0 | 54.9 | 47.9 | 54.6 | 80.9 | 75.9 | 78.0 |
| DeepSeek-Coder-V2-Lite-Instruct | **46.3** | **48.4** | **58.2** | **56.0** | **57.1** | **51.1** | **55.6** | 82.0 | 74.7 | 77.9 |
| DeepSeek-Coder-6.7B-Instruct | 43.0 | 45.4 | 56.5 | 55.6 | 56.0 | 46.8 | 52.9 | 81.3 | **77.5** | **78.7** |
| ReflectionCoder-DS-6.7B | 43.4 | 45.4 | 55.7 | 55.1 | 55.2 | 46.7 | 51.6 | 79.3 | 74.8 | 75.9 |
| CodeQwen1.5-7B-Chat | 45.8 | 47.4 | 56.3 | 55.6 | 56.3 | 49.1 | 53.2 | 78.0 | 74.1 | 76.3 |
| StarCoder2-15B-Instruct-v0.1 | 43.1 | 44.2 | 54.1 | 53.0 | 53.3 | 45.8 | 49.0 | 78.0 | 72.2 | 72.7 |
| CodeLlama-13B-Instruct | 34.8 | 36.9 | 47.2 | 46.9 | 47.2 | 37.8 | 43.2 | 73.1 | 68.9 | 68.9 |

## 5.3 RESULTS ON CONVCODEBENCH

While CONVCODEWORLD provides valuable insights into interactive code generation across various feedback combinations, CONVCODEBENCH offers a faster, cheaper, and more reproducible alternative. As discussed in §4, we chose CodeLlama-7B-Instruct as the reference model, and excluded scenarios without verbal feedback, as they do not require LLM intervention. Additionally, $\langle f_c, \phi, f_v \rangle$ scenario was omitted as CodeLlama-7B-Instruct achieved a 100% compilation success rate in the initial generation, eliminating the need for novice-level verbal feedback on compilation.

**CONVCODEBENCH as a Reliable Proxy** We conducted a comparative analysis of CONVCODEBENCH and CONVCODEWORLD to validate CONVCODEBENCH as a proxy, comparing the MRR (Figure 2) and Recall (Appendix H.1) results across target models and feedback combinations Spearman's rank correlations ranged from 0.82–0.99, indicating that CONVCODEBENCH is a reliable, efficient, and cost-effective proxy for CONVCODEWORLD.

Additionally, Table 6 presents the results on CONVCODEBENCH, showing that MRR ranking trends closely aligned with CONVCODEWORLD (Table 4), with minor deviations. While absolute recall and MRR scores are slightly lower compared to CONVCODEWORLD, the rankings amongst models remained roughly consistent between CONVCODEBENCH and CONVCODEWORLD. Based on approximately consistent rankings across CONVCODEWORLD and CONVCODEBENCH, **we recommend code LLMs use CONVCODEBENCH as a solid alternative to compare against other baselines.**

## 6 CONCLUSION

This paper recognizes the need for benchmarks with diverse type of interactions in conversational code generation. To address this gap, we introduced CONVCODEWORLD, a novel and reproducible environment designed to assess LLM code generation abilities across nine varied feedback scenarios. Additionally, for scenarios where API call costs are prohibitive, we offer CONVCODEBENCH, a zero-call benchmark from pre-generated feedback logs, providing a highly correlated evaluation of the conversational code generation capabilities of LLMs with CONVCODEWORLD. Our work contributes to a more thorough evaluation of diverse multi-turn evaluation objectives, and highlights a gap to invite for future models in the new design space.

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

# A  DEEKSEEK-R1-DISTILL RESULTS ON CONVCODEWORLD

Table 7: MRR results on CONVCODEWORLD. ✗ indicates that no feedback of that type is provided ($\phi$). The leftmost results, with three ✗, represent $\Omega = \langle \phi, \phi, \phi \rangle$, corresponding to single-turn code generation without any feedback. For each column, bold and underscore indicate 1st and 2nd place performance within the same model group. Maximum token length is set to 8K throughout the experiments, except for the R1-Distill models, which are set to 16K.

| Compilation Feedback | ✗ | $f_c$ | $f_c$ | $f_c$ | $f_c$ | $f_c$ | $f_c$ | $f_c$ | $f_c$ | $f_c$ |
|---|---|---|---|---|---|---|---|---|---|---|
| Execution Feedback | ✗ | ✗ | $f_e$ | $f_e^*$ | ✗ | $f_e$ | $f_e^*$ | ✗ | $f_e$ | $f_e^*$ |
| Verbal Feedback | ✗ | ✗ | ✗ | ✗ | $f_v$ | $f_v$ | $f_v$ | $f_v^*$ | $f_v^*$ | $f_v^*$ |
| **Closed-Source Models** | | | | | | | | | | |
| GPT-4-0613 | 46.0 | 46.0 | 52.1 | 56.1 | 46.0 | 52.4 | 56.4 | 63.1 | 64.3 | 64.8 |
| GPT-4-Turbo-2024-04-09 | 48.0 | 48.0 | 51.8 | 54.8 | 48.0 | 52.6 | 56.4 | 62.4 | 64.3 | 64.5 |
| GPT-4o-2024-05-13 | **50.8** | **50.8** | 55.0 | 57.9 | **50.8** | 55.1 | 58.6 | 63.3 | 64.7 | 65.3 |
| **Open-Source Models (≥ 30B)** | | | | | | | | | | |
| DeepSeek-R1-Distill-Llama-70B (16K) | 46.1 | 46.2 | 51.7 | 55.2 | 46.2 | 51.3 | 55.3 | 58.0 | 59.5 | 59.7 |
| Llama-3.3-70B-Instruct | **47.6** | **47.7** | 52.6 | 56.0 | **47.7** | 53.3 | 57.0 | **61.6** | 63.9 | **64.1** |
| DeepSeek-R1-Distill-Qwen-32B (16K) | 45.9 | 45.9 | 51.2 | 54.3 | 45.9 | 51.7 | 55.8 | 60.3 | 61.4 | 62.4 |
| Qwen2.5-32B | 45.8 | 45.8 | 47.9 | 49.5 | 45.8 | 49.8 | 53.4 | **61.6** | 62.7 | 63.8 |
| Llama-3.1-70B-Instruct | 45.4 | 45.4 | 49.9 | 53.4 | 45.4 | 50.8 | 55.2 | 60.7 | 62.6 | 63.3 |
| DeepSeek-Coder-33B-Instruct | 41.6 | 41.6 | 43.4 | 43.6 | 41.6 | 45.5 | 48.0 | 58.6 | 58.5 | 58.8 |
| ReflectionCoder-DS-33B | 41.6 | 41.6 | 42.9 | 42.9 | 41.6 | 45.6 | 48.1 | 57.7 | 58.2 | 58.91 |
| Qwen1.5-72B-Chat | 32.9 | 33.0 | 35.8 | 38.3 | 33.0 | 38.6 | 41.4 | 50.6 | 52.0 | 52.7 |
| Qwen1.5-32B-Chat | 32.0 | 32.0 | 35.3 | 36.7 | 32.0 | 36.6 | 39.7 | 47.4 | 42.6 | 40.8 |
| CodeLlama-34B-Instruct | 28.8 | 28.8 | 31.0 | 31.9 | 28.8 | 32.5 | 35.1 | 48.7 | 49.2 | 49.8 |
| **Open-Source Models (< 30B)** | | | | | | | | | | |
| Llama-3.1-8B-Instruct | 31.4 | 31.5 | 34.0 | 34.6 | 31.5 | 36.1 | 39.1 | 49.4 | 49.8 | 51.3 |
| DeepSeek-Coder-V2-Lite-Instruct | 38.3 | 38.3 | **40.5** | **41.7** | 38.3 | **42.0** | **43.8** | 52.7 | 52.9 | 53.3 |
| DeepSeek-Coder-6.7B-Instruct | 35.2 | 35.2 | 36.2 | 36.1 | 35.2 | 38.8 | 40.5 | 53.3 | 53.2 | 53.9 |
| ReflectionCoder-DS-6.7B | 37.4 | 37.4 | 38.3 | 38.7 | 37.4 | 40.4 | 42.4 | 53.3 | **53.8** | 53.6 |
| CodeQwen1.5-7B-Chat | **39.3** | **39.4** | 39.7 | 40.1 | **39.3** | **42.0** | 43.7 | 53.7 | 53.5 | **54.8** |
| StarCoder2-15B-Instruct-v0.1 | 37.1 | 37.1 | 37.9 | 38.3 | 37.1 | 39.4 | 40.5 | 52.7 | 52.8 | 52.1 |
| CodeLlama-13B-Instruct | 28.4 | 28.4 | 29.0 | 29.0 | 28.4 | 31.2 | 33.0 | 43.9 | 44.3 | 44.8 |
| CodeLlama-7B-Instruct | 21.8 | 21.8 | 22.3 | 22.3 | 21.8 | 23.5 | 25.2 | 35.0 | 33.4 | 33.9 |

Table 8: Recall results on CONVCODEWORLD. ✗ indicates that no feedback of that type is provided ($\phi$). The leftmost results, with three ✗, represent $\Omega = \langle \phi, \phi, \phi \rangle$, corresponding to single-turn code generation without any feedback. For each column, bold and underscore indicate 1st and 2nd place performance within the same model group. Maximum token length is set to 8K throughout the experiments, except for the R1-Distill models, which are set to 16K.

| Compilation Feedback | ✗ | $f_c$ | $f_c$ | $f_c$ | $f_c$ | $f_c$ | $f_c$ | $f_c$ | $f_c$ | $f_c$ |
|---|---|---|---|---|---|---|---|---|---|---|
| Execution Feedback | ✗ | ✗ | $f_e$ | $f_e^*$ | ✗ | $f_e$ | $f_e^*$ | ✗ | $f_e$ | $f_e^*$ |
| Verbal Feedback | ✗ | ✗ | ✗ | ✗ | $f_v$ | $f_v$ | $f_v$ | $f_v^*$ | $f_v^*$ | $f_v^*$ |
| **Closed-Source Models** | | | | | | | | | | |
| GPT-4-0613 | 46.0 | 46.0 | 60.3 | **70.5** | 46.0 | **61.9** | **72.5** | **89.7** | **91.1** | **92.5** |
| GPT-4-Turbo-2024-04-09 | 48.0 | 48.0 | 56.7 | 63.8 | 48.0 | 58.6 | 68.1 | 84.7 | 87.5 | 88.5 |
| GPT-4o-2024-05-13 | **50.8** | **50.8** | 60.5 | 67.6 | **50.8** | 60.8 | 69.6 | 82.3 | 84.9 | 86.2 |
| **Open-Source Models (≥ 30B)** | | | | | | | | | | |
| DeepSeek-R1-Distill-Llama-70B (16K) | 46.1 | 46.2 | **61.7** | **72.7** | 46.2 | 60.2 | 73.8 | 82.0 | 86.8 | 86.1 |
| Llama-3.3-70B-Instruct | **47.6** | **47.7** | 59.0 | 67.7 | **47.7** | 61.5 | 72.2 | 84.6 | 87.6 | 88.9 |
| DeepSeek-R1-Distill-Qwen-32B (16K) | 45.9 | 45.9 | 59.5 | 68.1 | 45.9 | 61.2 | **74.0** | 85.0 | 88.1 | 89.0 |
| Qwen2.5-32B | 45.8 | 45.9 | 50.4 | 53.9 | 46.0 | 54.8 | 62.6 | 84.7 | 85.5 | 87.5 |
| Llama-3.1-70B-Instruct | 45.4 | 45.4 | 56.2 | 64.8 | 45.4 | 59.5 | 70.8 | **86.7** | **88.9** | **91.8** |
| DeepSeek-Coder-33B-Instruct | 41.6 | 41.6 | 45.5 | 46.1 | 41.6 | 50.4 | 56.6 | 85.4 | 84.6 | 85.6 |
| ReflectionCoder-DS-33B | 41.6 | 41.6 | 45.3 | 44.9 | 41.6 | 51.4 | 57.2 | 81.4 | 81.8 | 84.2 |
| Qwen1.5-72B-Chat | 32.9 | 33.2 | 39.9 | 47.5 | 33.2 | 47.5 | 57.9 | 84.4 | 86.1 | 87.2 |
| Qwen1.5-32B-Chat | 32.0 | 32.0 | 41.1 | 45.3 | 32.0 | 44.6 | 54.3 | 75.9 | 61.8 | 57.1 |
| CodeLlama-34B-Instruct | 28.8 | 28.8 | 33.7 | 35.8 | 28.8 | 37.5 | 44.6 | 80.0 | 82.0 | 82.3 |
| **Open-Source Models (< 30B)** | | | | | | | | | | |
| Llama-3.1-8B-Instruct | 31.4 | 31.8 | 38.4 | 40.0 | 31.7 | 43.2 | **51.8** | 80.9 | 80.2 | **83.7** |
| DeepSeek-Coder-V2-Lite-Instruct | 38.3 | 38.3 | **43.4** | **46.1** | 38.3 | **47.0** | 51.4 | 76.3 | 75.8 | 76.9 |
| DeepSeek-Coder-6.7B-Instruct | 35.2 | 35.2 | 37.7 | 37.5 | 35.2 | 43.3 | 48.2 | **82.8** | **82.5** | 83.1 |
| ReflectionCoder-DS-6.7B | 37.4 | 37.4 | 39.6 | 40.7 | 37.4 | 44.7 | 50.4 | 79.1 | 79.6 | 78.9 |
| CodeQwen1.5-7B-Chat | **39.3** | **39.6** | 40.1 | 41.1 | **39.5** | 45.8 | 49.5 | 74.4 | 74.7 | 77.4 |
| StarCoder2-15B-Instruct-v0.1 | 37.1 | 37.1 | 39.3 | 40.0 | 37.1 | 42.6 | 46.3 | 76.9 | 76.8 | 75.6 |
| CodeLlama-13B-Instruct | 28.4 | 28.4 | 29.7 | 30.0 | 28.4 | 35.1 | 41.1 | 69.0 | 70.7 | 71.6 |
| CodeLlama-7B-Instruct | 21.8 | 21.8 | 22.9 | 23.0 | 21.8 | 26.2 | 30.5 | 61.7 | 53.9 | 55.2 |

Table 9: Hyperparameter tuning results of DeepSeek-R1-Distill-Qwen-32B on BigCodeBench-Hard-Instruct.

| | Temperature | Max Token Length | Pass@1 |
|---|---|---|---|
| Reported | - | - | 43.9 |
| | 0.0 | 8K | 43.7 |
| | 0.2 | 8K | 44.6 |
| Reproduced | 0.2 | 16K | **45.9** |
| | 1.0 | 8K | 44.6 |
| | 1.0 | 16K | 45.1 |

We present the CONVCODEWORLD results for the DeepSeek-R1-Distill (DeepSeek-AI et al., 2025) models. These models, which are trained to handle more complex reasoning, required the following hyperparameter adjustments for inference:

- **Increased Token Length**: The maximum token length was increased from 8K to 16K (see Table 9) to support longer reasoning steps.

- **Temperature Adjustment**: The temperature was changed from 0.0 to 0.2. The 0.0 setting resulted in degeneration, causing repetitive sentences in reasoning. We also experimented with a temperature of 1.0, as o1 models only support this value,[6] but 0.2 provided the best performance.

Tables 7 and 8 extend the results presented in Tables 4 and 5, including two DeepSeek-R1-Distill models (Llama-70B and Qwen-32B) and their base models (Llama-3.3-70B-Instruct and Qwen2.5-32B). We summarize the key observations on the impact of R1-Distillation:

- **Lack of Significant Improvement**: R1-Distilled models do not demonstrate a significant improvement over other models.

- **Reduced Expert Feedback Utilization**: R1-Distilled models face challenges in effectively utilizing expert feedback over their base models.

- **DeepSeek-R1-Distill-Llama-70B vs. Llama-3.3-70B-Instruct**:
  - **MRR**: R1-Distillation results in a decrease in MRR performance.
  - **Recall**: R1-Distillation generally improves the utilization of execution and novice feedback but hurts expert feedback.

- **DeepSeek-R1-Distill-Qwen-32B vs. Qwen2.5-32B**:
  - **MRR**: R1-Distillation improves the utilization of execution and novice feedback but slightly hurts expert feedback.
  - **Recall**: R1-Distillation improves feedback utilization across all types of feedback in most feedback combinations.

## B    DISTINCTION OF CONVCODEWORLD

We elaborate distinctive implications from existing works such as InterCode Yang et al. (2023) and MINT Wang et al. (2024):

- Comparative analyses of partial to full test coverage in execution feedback enables to evaluate both:
  - **Test generalization:** A model's ability to produce code that passes full tests even when only partial tests are provided.
  - **Test utilization:** A model's capability to leverage given test results for code refinement.

  InterCode—-which uses full test only—-evaluates test utilization only, and MINT—-which uses partial test only—-provides an entangled evaluation of test generalization and test

---

[6]https://community.openai.com/t/why-is-the-temperature-and-top-p-of-o1-models-fixed-to-1-not-0/938922

utilization. In contrast, CONVCODEWORLD, by providing both partial and full test, enables **isolated evaluation of each test generalization and test utilization** as we illustrate below. For instance, in Table 5, DeepSeek-Coder-6.7B-Instruct shows modest test generalization ($\langle f_c, \phi, \phi \rangle \to \langle f_c, f_e, \phi \rangle$: 35.2 → 37.7), but shows limited test utilization ($\langle f_c, f_e, \phi \rangle \to \langle f_c, f_e^*, \phi \rangle$: 37.7 → 37.5). Meanwhile, Llama-3.1-8B-Instruct exhibits capabilities in both test generalization ($\langle f_c, \phi, \phi \rangle \to \langle f_c, f_e, \phi \rangle$: 31.8 → 38.4) and test utilization ($\langle f_c, f_e, \phi \rangle \to \langle f_c, f_e^*, \phi \rangle$: 38.4 → 40.0).

- CONVCODEWORLD simulates an **"engaged"** user, offering verbalized explanations of test results, as illustrated in Figure 17. In contrast, InterCode lacks verbal feedback, and MINT provides only generic feedback for $f_v$—*"Your answer is wrong."* To evaluate how verbalized explanations enhance models' test utilization capabilities, an exemplar scenario in CONVCODEWORLD is when both full execution feedback and novice feedback are available. In Table 5:

  - Without the inclusion of novice feedback ($\langle f_c, f_e^*, \phi \rangle$): Llama-3.1-8B-Instruct's test utilization capabilities (40.0) are weaker compared to CodeQwen1.5-7B-Chat (**41.1**).
  - With the inclusion of novice feedback ($\langle f_c, f_e^*, f_v \rangle$): significantly improves Llama-3.1-8B-Instruct's performance, surpassing CodeQwen1.5-7B-Chat (**51.8** vs. 49.5).

- Covering comprehensive combinations of feedback types, CONVCODEWORLD analyzes previously underexplored cases, such as:

  - Full execution feedback vs. partial execution and novice feedback
  - Partial execution and expert feedback vs. full execution and expert feedback
  - Full execution and novice feedback vs. expert feedback

- Cost-effective static benchmark (CONVCODEBENCH): CONVCODEBENCH correlates strongly with online evaluation while reducing costs. Neither MINT nor InterCode provide such a static benchmark.

## C  VERBAL FEEDBACK

### C.1  DISCUSSION ON EMPLOYING LLMS FOR VERBAL FEEDBACK GENERATION

A key challenge in creating CONVCODEWORLD is generating verbal feedback. Human annotation is both impractical and inconsistent (§3.1.1), which led us to employ GPT-4o for this task. While GPT-4o may not fully replicate the nuances of human feedback, it ensures reproducibility and affordability, both critical for maintaining consistency across benchmark evaluations. As demonstrated by direct comparisons between LLM-generated and human feedback in prior studies (Wang et al., 2024), we find this method sufficiently effective for our benchmarking purposes.

### C.2  COST-EFFICIENCY OF CONVCODEWORLD COMPARED TO HUMAN ANNOTATION

In the worst-case scenario, CodeLlama-7B-Instruct, which requested the most verbal feedback due to its low performance, incurred a total cost of $215 (26.4M input tokens and 5.5M output tokens) for 15,905 turns using GPT-4o-2024-05-13 pricing ($5/1M input tokens and $15/1M output tokens). By comparison, assuming human annotation takes 96 seconds per turn (Wang et al., 2024) and the average U.S. private non-farmer worker's hourly wage is $35.04 according to US Bureau of Labor Statistics (2024), the human annotation cost would be approximately $14,792.

### C.3  HUMAN EVALUATION OF GENERATED VERBAL FEEDBACK

Table 10: Human evaluation of simulated expert-level user feedback by GPT-4o and real user feedback by ShareGPT.

| Expert Feedback by | Is Helpful | Is Human-Expert-Like |
|---|---|---|
| ShareGPT | 35% | 30% |
| CONVCODEWORLD | 55% | 25% |

We conducted human evaluation to validate the realism of simulated expert-level user feedback, noting that in-context examples might lead to unrealistic responses. Specifically, two human evaluators

rated randomly assigned feedback samples from either real user feedback from ShareGPT [7] logs or expert feedback generated by CONVCODEWORLD using GPT-4o (see Figure 19 for the annotation platform). As shown in Table 10, our generated feedback was found to be comparable to authentic logs in terms of expert-human-likeness and was rated higher for helpfulness, consistent with MINT's findings.

## C.4 POSSIBLE REASONS FOR THE OBSERVED "STRUGGLE TO UTILIZE FEEDBACK"

From Section 5.2.1, we further discuss two possible reasons for models when they struggle to utilize complex feedback:

- **Limited Model Size**: Smaller models, such as ReflectionCoder-DS-6.7B, may lack the capacity to process and integrate complex information effectively, which could limit performance improvements even when execution feedback is included ($35.2 \rightarrow 37.7$). In contrast, their bigger versions like ReflectionCoder-DS-33B demonstrated performance gains with execution feedback ($41.6 \rightarrow 45.3$). Mixed feedback types may distract small models further. When comparing Expert feedback only vs. Expert feedback + execution feedback. For Qwen1.5-Chat, the 72B model's performance improved with execution feedback, while the 32B model's performance deteriorated, which suggests that smaller models might become distracted when faced with multiple feedback signals simultaneously (Liu et al., 2024). However, this distraction may be mitigated with well-designed training data, as even smaller models like Llama-3.1-8B-Instruct show improvements when provided with more execution feedback.

- **Limited Generalization Training**: ReflectionCoder models were trained on a specific feedback combination, $\langle f_c, f_e^*, f_v \rangle$, limiting their adaptability to other feedback types (Section 5.2.3). For example, with expert feedback, ReflectionCoder-DS-33B scores lower (81.4) than its base model DeepSeekCoder-33B-Instruct (85.4).

## C.5 ANALYSIS OF GROUND TRUTH CODE LEAKAGE IN GENERATED EXPERT-LEVEL VERBAL FEEDBACK

Table 11: Pass@1 results of various LLMs with expert-level verbal feedback $f_v^*$ generated by GPT-4o compared to direct ground truth code feedback. The total number of turns $n = 1$. For each column, bold and underscore indicate 1st and 2nd place performance while keeping the code generation model fixed.

| Feedback | Code Generation | | |
|---|---|---|---|
| | GPT-4-0613 | GPT-4-Turbo-2024-04-09 | GPT-4o-2024-05-13 |
| **w/o Feedback** | 46.0 | 48.0 | 50.8 |
| **+ Expert-Level Verbal Feedback** | 70.0 | 69.0 | 68.5 |
| **+ Ground Truth Code** | **97.9** | **88.2** | **79.7** |

Table 12: Ground truth code leakage ratio (%) by incorporating different models for expert-level verbal feedback generation. The lower the better.

| $f_v^*$ Generation | Mentioning ground_truth_code ($\downarrow$) | Including Refined Code ($\downarrow$) |
|---|---|---|
| GPT-4-0613 | 51.1 | 0.0 |
| GPT-4-Turbo-2024-04-09 | 31.4 | 0.0 |
| GPT-4o-2024-05-13 | 2.5 | 0.1 |

The generation of expert-level verbal feedback $f_v^*$ involves comparing the generated code with the ground truth code to provide modification suggestions. This process could raise concerns about potential code leakage. As shown in Table 11, providing the ground truth code significantly outperforms providing $f_v^*$, empirically confirming that $f_v^*$ is unlikely to be a direct copy of the ground truth code.

---

[7] https://huggingface.co/datasets/anon8231489123/ShareGPT_Vicuna_unfiltered/blob/main/ShareGPT_V3_unfiltered_cleaned_split_no_imsorry.json

To detect leakage, we use a **canary sequence** approach, commonly used to test for training data or prompt leakage in LLMs (Team et al., 2024; OpenAI, 2023; Greshake et al., 2023; Perez & Ribeiro, 2022; Agarwal et al., 2024). Specifically, we consider leakage if the feedback simulator includes a canary sequence within the feedback. This sequence contains the term *ground truth code*, which is given in the prompt (see Figure 15). As shown in Table 12, leakage rates are estimated by how often a model references the ground truth code in $f_v^*$. For example, a leakage might be detected if the feedback contains phrases such as, *"Unlike the ground truth code, the current code omits exception handling of DivideByZero..."* (see Figures 20 and 21 for comparisons of desirable vs. leaked cases).

Notably, GPT-4o shows the lowest leakage rate at 2.5%, indicating its ability to generate $f_v^*$ with minimal leakage. This suggests that when $f_v^*$ generated by GPT-4o is provided, the observed performance improvement is not driven by exposure to the correct code.

### C.6 COMPARATIVE ANALYSIS OF VERBAL FEEDBACK ACROSS DIFFERENT LLMS

Table 13: Pass@1 results over different model combinations of expert-level verbal feedback $f_v^*$ generation and code generation on CONVCODEWORLD where $\Omega = \langle f_c, \phi, f_v^* \rangle$ and the total number of turns $n = 1$. Each row represents a model used to provide verbal feedback. Each column represents a model that utilizes this feedback to refine code. For each column, bold and underscore indicate 1st and 2nd place performance while keeping the code generation model fixed.

| $f_v^*$ Generation | Code Generation | | |
| --- | --- | --- | --- |
| | GPT-4-0613 | GPT-4-Turbo-2024-04-09 | GPT-4o-2024-05-13 |
| w/o Feedback | 46.0 | 48.0 | 50.8 |
| GPT-4-0613 | 65.1 | 61.4 | 63.4 |
| GPT-4-Turbo-2024-04-09 | 62.9 | 59.9 | 62.5 |
| GPT-4o-2024-05-13 | **67.1** | **65.4** | **64.2** |

In our main experiments, we utilized GPT-4o for verbal feedback generation and investigated its performance in comparison to other models. To see the effect of using other LLMs for verbal feedback generation, we conducted a single iteration of code generation using three closed-source LLMs as both code generators and expert-level verbal feedback generators, examining the Pass@1 performance. Table 13 evaluates different models as potential verbal feedback simulators. The effectiveness of the feedback provided by each simulator is assessed by comparing the performance across columns, showing consistent superior performance when employing GPT-4o for feedback generation.

## D VERBAL FEEDBACK BY OPEN-SOURCE LLMS

Table 14: Pass@1 results over different model combinations of expert-level verbal feedback $f_v^*$ generation and code generation on CONVCODEWORLD where $\Omega = \langle f_c, \phi, f_v^* \rangle$ and the total number of turns $n = 1$. For each column, bold and underscore indicate 1st and 2nd place performance while keeping the code generation model fixed.

| $f_v^*$ Generation | Code Generation | |
| --- | --- | --- |
| | GPT-4o-2024-05-13 | Llama-3.1-70B-Instruct |
| w/o Feedback | 50.8 | 45.4 |
| GPT-4o-2024-05-13 | 64.2 | **65.1** |
| Llama-3.1-70B-Instruct | **65.8** | 62.1 |

Table 14 supports the feasibility of using Llama-3.1-70B-Instruct as a verbal feedback simulator, replacing GPT-4o-2024-05-13.

## E IMPLEMENTATION DETAILS

We utilize DSPy (Khattab et al., 2024)[8] manage the interactive code generation flow for CONVCODEWORLD and CONVCODEBENCH. For both code and verbal feedback generation follow DSPy's default prompt format, incorporating ChaingOfThought (CoT) (Wei et al., 2022)

---

[8] https://github.com/stanfordnlp/dspy

Table 15: Pass@1 results over different implementation for initial code generation without feedback. CONVCODEWORLD chose Direct Generation by BigCodeBench implementation, which showed the highest performance. For each column, bold and underscore indicate 1st and 2nd place performance while keeping the code generation model fixed.

| Implementation | DeepSeek-Coder-6.7B-Instruct | GPT-4o-2024-05-13 |
|---|---|---|
| **w/o Feedback** ($\Omega = \langle \phi, \phi, \phi \rangle$) | | |
| Reported | 35.5 | 51.1 |
| Direct Generation (BigCodeBench impl.) | **35.2** | **50.8** |
| DSPy.Predict | 33.6 | 1.8 |
| DSPy.ChainOfThought | 20.2 | 49.3 |
| **Compilation Feedback only** ($\Omega = \langle f_c, \phi, \phi \rangle; n = 1$) | | |
| Direct Generation (BigCodeBench impl.) | **35.2** | **50.8** |
| DSPy.Predict | 33.7 | 50.1 |
| DSPy.ChainOfThought | 32.8 | 50.5 |

reasoning by `DSPy.ChainOfThought` function. The exception is initial code generation, where we adopt BigCodeBench's (Zhuo et al., 2024) implementation,[9] without CoT reasoning. As shown in Table 15, we attribute this choice to the observation that, for initial code generation (without prior feedback), models tend to perform better without additional reasoning steps like CoT (`DSPy.ChainOfThought`) or prompting (both in `DSPy.Predict` and `DSPy.ChainOfThought`).

Hyperparameters are set as follows: We used greedy decoding (temperature = 0) in all experiments, following Chen et al. (2023). The total number of turns $n = 10$, with a maximum token length of 8K for all code generation models. For models with a lower token limit, we use their respective maximum length. For verbal feedback generation, we use GPT-4o-2024-05-13 with a token limit of 2K. Regarding the partial test coverage of execution feedback, we utilize the first three test cases, which aligns with benchmarks like HumanEval (Chen et al., 2021) and CodeContests (Li et al., 2022) providing up to three public test cases.

---

[9]https://github.com/bigcode-project/bigcodebench

# F    CONVCODEWORLD

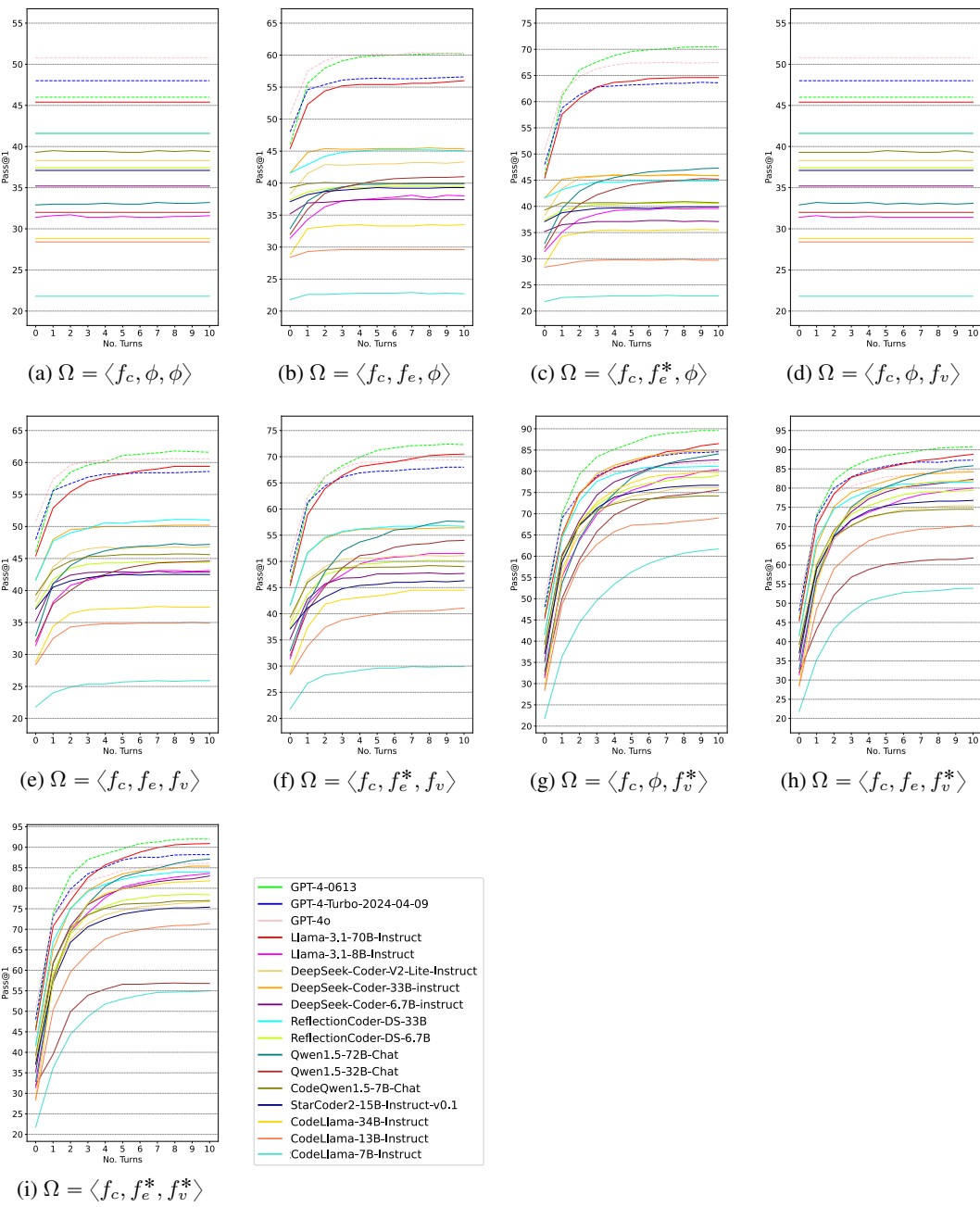

Figure 3: Iterative Pass@1 results on CONVCODEWORLD with different feedback combinations $\Omega$.

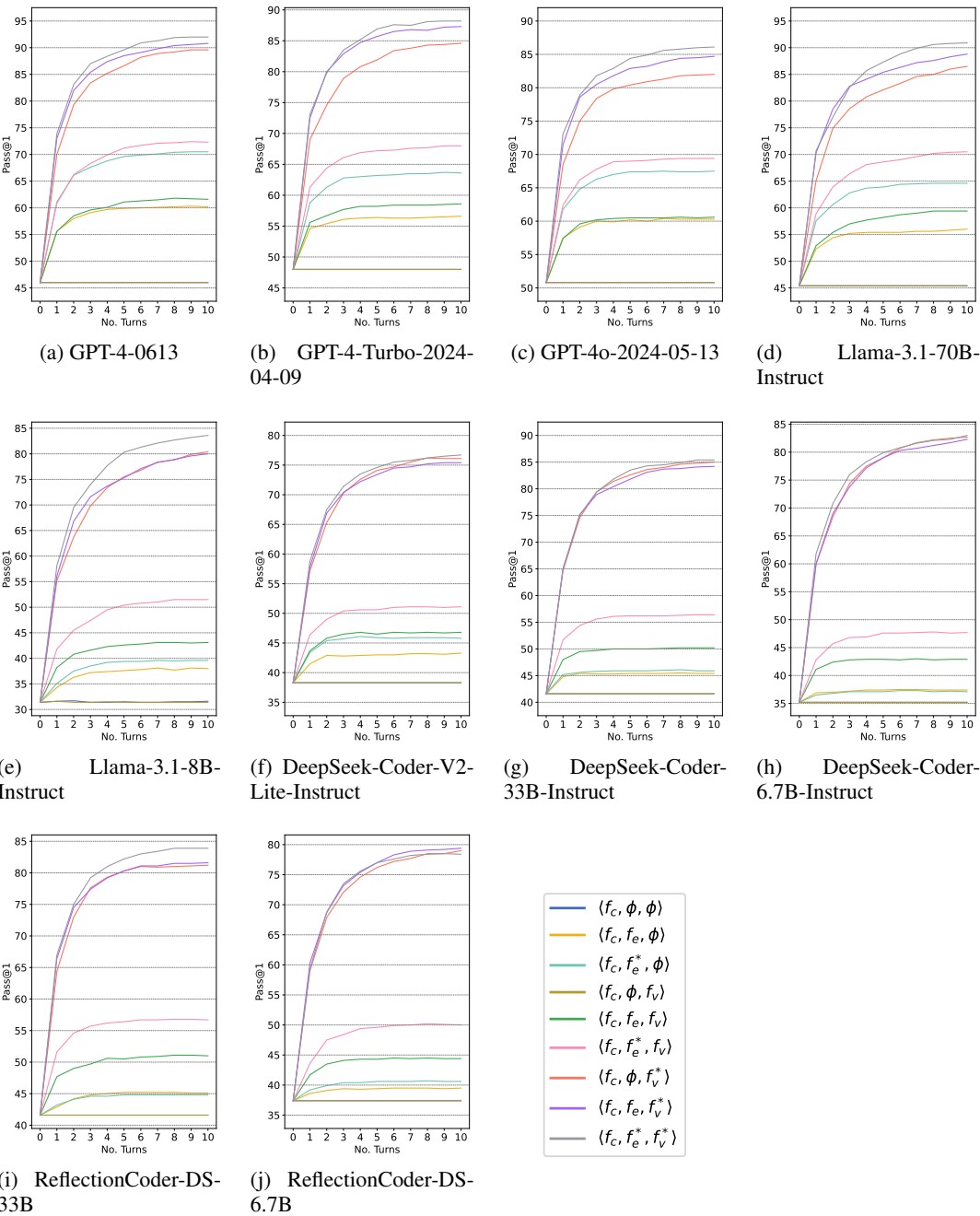

Figure 4: Iterative Pass@1 results of each LLM on CONVCODEWORLD with different feedback combinations $\Omega$ (continued on Figure 5).

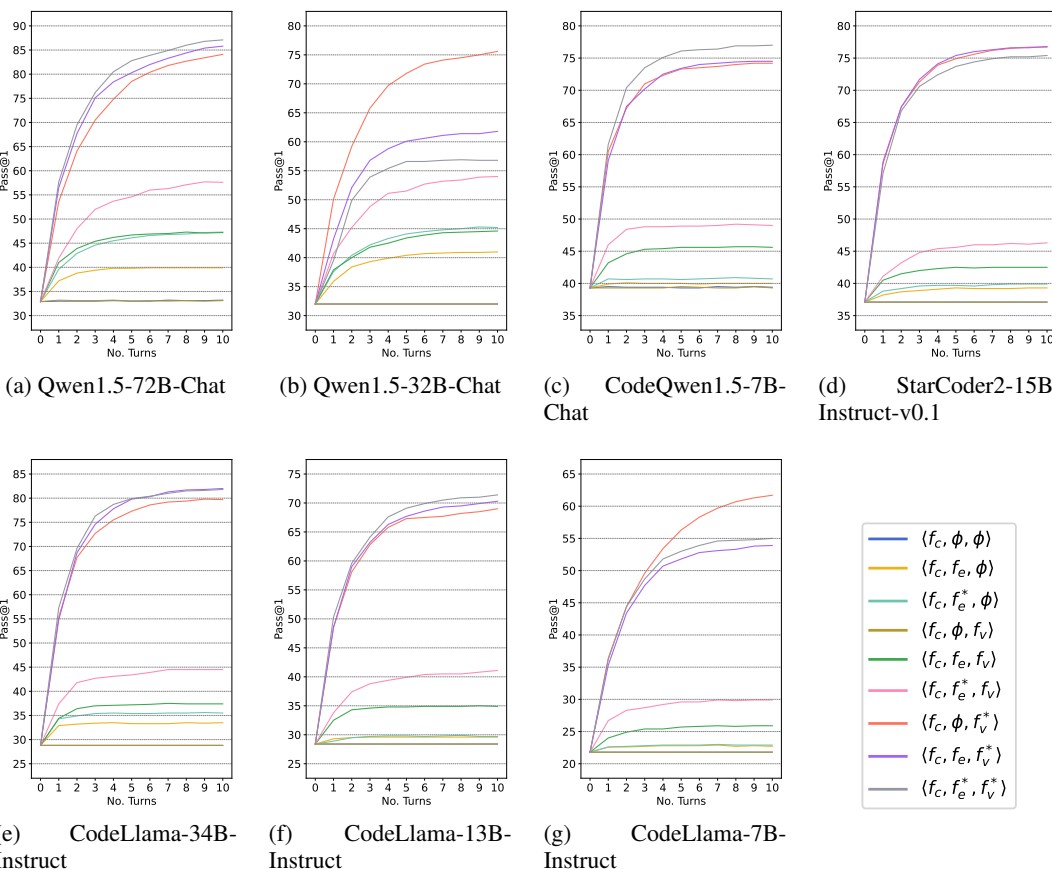

Figure 5: Iterative Pass@1 results of each LLM on CONVCODEWORLD with different feedback combinations $\Omega$ (continued from Figure 4).

# G CONVCODEBENCH

## G.1 MRR AND RECALL RESULTS

### G.1.1 REFERENCE MODEL: DEEPSEEK-CODER-6.7B-INSTRUCT

Table 16: MRR and Recall results on CONVCODEBENCH using logs of DeepSeek-Coder-6.7B-Instruct in CONVCODEWORLD. ✗ indicates that no feedback of that type is provided ($\phi$). For each column, bold and underscore indicate 1st and 2nd place performance within the same model group.

| | MRR | | | | | Recall | | | | |
|---|---|---|---|---|---|---|---|---|---|---|
| Compilation Feedback | $f_c$ | $f_c$ | $f_c$ | $f_c$ | $f_c$ | $f_c$ | $f_c$ | $f_c$ | $f_c$ | $f_c$ |
| Execution Feedback | $f_e$ | $f_e^*$ | ✗ | $f_e$ | $f_e^*$ | $f_e$ | $f_e^*$ | ✗ | $f_e$ | $f_e^*$ |
| Verbal Feedback | $f_n$ | $f_n$ | $f_n^*$ | $f_n^*$ | $f_n^*$ | $f_n$ | $f_n$ | $f_n^*$ | $f_n^*$ | $f_n^*$ |
| Closed-Source Models | | | | | | | | | | |
| GPT-4-0613 | 56.2 | 59.1 | 66.9 | 67.4 | 68.2 | 61.8 | **68.9** | **89.9** | **90.6** | **91.0** |
| GPT-4-Turbo-2024-04-09 | 57.4 | 60.1 | 67.6 | 68.3 | 69.0 | 61.7 | 68.3 | 89.0 | 89.9 | 90.0 |
| GPT-4o-2024-05-13 | **58.8** | **61.3** | **69.0** | **69.3** | **70.2** | **63.1** | **68.9** | 89.8 | 90.1 | 90.5 |
| Open-Source Models ($\geqslant$ 30B) | | | | | | | | | | |
| Llama-3.1-70B-Instruct | **57.2** | **59.2** | **67.2** | **67.7** | **68.5** | **62.3** | **67.0** | **89.4** | **89.7** | **90.4** |
| DeepSeek-Coder-33B-Instruct | 52.4 | 54.0 | 63.4 | 64.4 | 65.3 | 56.2 | 60.7 | 86.8 | 87.8 | 88.6 |
| ReflectionCoder-DS-33B | 52.6 | 54.7 | 64.0 | 64.5 | 65.3 | 56.4 | 62.0 | 86.8 | 87.8 | 88.2 |
| Qwen1.5-72B-Chat | 49.1 | 52.0 | 61.4 | 61.9 | 62.7 | 54.6 | 61.8 | 87.6 | 88.2 | **88.8** |
| Qwen1.5-32B-Chat | 48.6 | 50.8 | 60.4 | 59.9 | 60.1 | 54.1 | 59.2 | 86.3 | 84.8 | 84.8 |
| CodeLlama-34B-Instruct | 47.2 | 48.8 | 60.6 | 61.1 | 61.6 | 51.7 | 56.4 | 87.4 | 88.2 | 88.2 |
| Open-Source Models ($<$ 30B) | | | | | | | | | | |
| Llama-3.1-8B-Instruct | 50.6 | 52.5 | 62.3 | 62.8 | 63.4 | 55.8 | 61.2 | **87.3** | **88.3** | **88.2** |
| DeepSeek-Coder-V2-Lite-Instruct | **52.4** | **54.4** | **63.1** | **63.8** | **64.7** | **56.4** | **61.7** | 86.2 | 87.1 | 87.7 |
| ReflectionCoder-DS-6.7B | 48.5 | 50.2 | 61.0 | 61.2 | 61.8 | 52.5 | 56.9 | 85.8 | 85.9 | 86.4 |
| CodeQwen1.5-7B-Chat | 51.5 | 53.6 | 62.8 | 63.5 | 64.0 | 55.2 | 60.8 | 86.1 | 86.8 | 87.4 |
| StarCoder2-15B-Instruct-v0.1 | 49.7 | 51.7 | 62.3 | 62.2 | 62.8 | 52.9 | 58.1 | 86.6 | 85.9 | 86.6 |
| CodeLlama-13B-Instruct | 47.4 | 49.3 | 60.4 | 60.4 | 61.1 | 51.8 | 56.8 | 86.6 | 86.2 | 87.4 |
| CodeLlama-7B-Instruct | 44.2 | 45.7 | 57.9 | 57.9 | 58.3 | 48.9 | 53.2 | 86.3 | 86.1 | 85.4 |

### G.1.2 REFERENCE MODEL: GPT-4-0613

Table 17: MRR and Recall results on CONVCODEBENCH using logs of GPT-4-0613 in CONVCODEWORLD. ✗ indicates that no feedback of that type is provided ($\phi$). For each column, bold and underscore indicate 1st and 2nd place performance within the same model group.

| | MRR | | | | | Recall | | | | |
|---|---|---|---|---|---|---|---|---|---|---|
| Compilation Feedback | $f_c$ | $f_c$ | $f_c$ | $f_c$ | $f_c$ | $f_c$ | $f_c$ | $f_c$ | $f_c$ | $f_c$ |
| Execution Feedback | $f_e$ | $f_e^*$ | ✗ | $f_e$ | $f_e^*$ | $f_e$ | $f_e^*$ | ✗ | $f_e$ | $f_e^*$ |
| Verbal Feedback | $f_v$ | $f_v$ | $f_v^*$ | $f_v^*$ | $f_v^*$ | $f_v$ | $f_v$ | $f_v^*$ | $f_v^*$ | $f_v^*$ |
| Closed-Source Models | | | | | | | | | | |
| GPT-4-Turbo-2024-04-09 | 60.3 | 64.1 | 69.9 | 70.9 | 71.6 | 67.2 | 76.7 | 91.6 | 92.8 | 94.2 |
| GPT-4o-2024-05-13 | **61.6** | **65.0** | **70.6** | **71.5** | **72.3** | **68.6** | **77.2** | **91.9** | **93.0** | **94.3** |
| Open-Source Models ($\geqslant$ 30B) | | | | | | | | | | |
| Llama-3.1-70B-Instruct | **60.9** | **64.2** | **69.9** | **70.9** | **71.5** | **68.8** | **77.7** | **92.2** | **93.5** | **94.6** |
| DeepSeek-Coder-33B-Instruct | 58.3 | 61.9 | 68.2 | 69.3 | 69.9 | 66.5 | 75.9 | 91.9 | 93.2 | 94.3 |
| ReflectionCoder-DS-33B | 58.9 | 62.4 | 68.8 | 70.0 | 70.3 | 66.5 | 75.9 | 91.8 | 93.3 | 94.5 |
| Qwen1.5-72B-Chat | 57.5 | 60.4 | 67.3 | 68.3 | 69.1 | 66.0 | 73.9 | 91.5 | 92.5 | 94.2 |
| Qwen1.5-32B-Chat | 56.6 | 60.6 | 66.8 | 67.6 | 67.7 | 65.4 | 75.7 | 91.4 | 92.7 | 92.9 |
| CodeLlama-34B-Instruct | 56.2 | 59.9 | 66.8 | 67.8 | 68.4 | 64.7 | 74.8 | **92.2** | 93.1 | 94.4 |
| Open-Source Models ($<$ 30B) | | | | | | | | | | |
| Llama-3.1-8B-Instruct | 56.9 | 60.6 | 67.4 | 68.3 | 68.9 | 65.4 | 74.8 | 91.8 | 92.8 | 94.3 |
| DeepSeek-Coder-V2-Lite-Instruct | 58.8 | **62.4** | **68.9** | **69.7** | 70.1 | 66.4 | 75.5 | 91.8 | 92.6 | 93.9 |
| DeepSeek-Coder-6.7B-Instruct | 57.5 | 61.1 | 67.4 | 68.7 | 69.2 | 65.7 | 75.5 | **93.1** | **94.4** | 94.4 |
| ReflectionCoder-DS-6.7B | 57.9 | 61.5 | 68.0 | 69.1 | 69.7 | 65.7 | 75.2 | 91.9 | 93.0 | 94.1 |
| CodeQwen1.5-7B-Chat | **59.0** | **62.4** | 68.5 | 69.6 | **70.2** | **67.1** | **76.1** | 91.8 | 92.9 | **94.4** |
| StarCoder2-15B-Instruct-v0.1 | 58.3 | 61.8 | 68.0 | 68.9 | 69.7 | 66.0 | 75.3 | 91.2 | 92.5 | 94.0 |
| CodeLlama-13B-Instruct | 56.1 | 59.9 | 66.4 | 67.5 | 68.1 | 64.9 | 74.6 | 91.5 | 92.6 | **94.4** |
| CodeLlama-7B-Instruct | 54.8 | 58.4 | 65.5 | 66.4 | 67.0 | 63.7 | 73.4 | 91.9 | 92.5 | 93.6 |

# H RANK CORRELATIONS BETWEEN CONVCODEBENCH AND CONVCODEWORLD

## H.1 REFERENCE MODEL: CODELLAMA-7B-INSTRUCT-HF

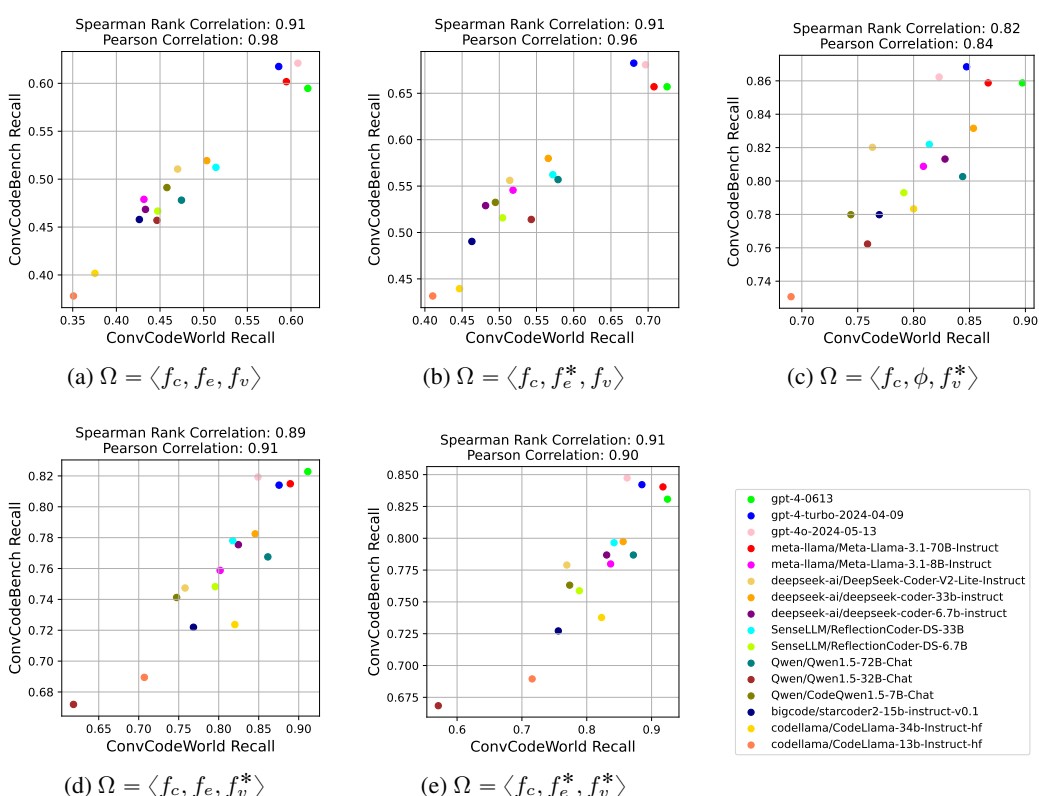

Figure 6: Correlation between Recall on CONVCODEBENCH (ref. CodeLlama-7B-Instruct) and Recall on CONVCODEWORLD with different feedback combinations $\Omega$.

### H.1.1  REFERENCE MODEL: DEEPSEEK-CODER-6.7B-INSTRUCT

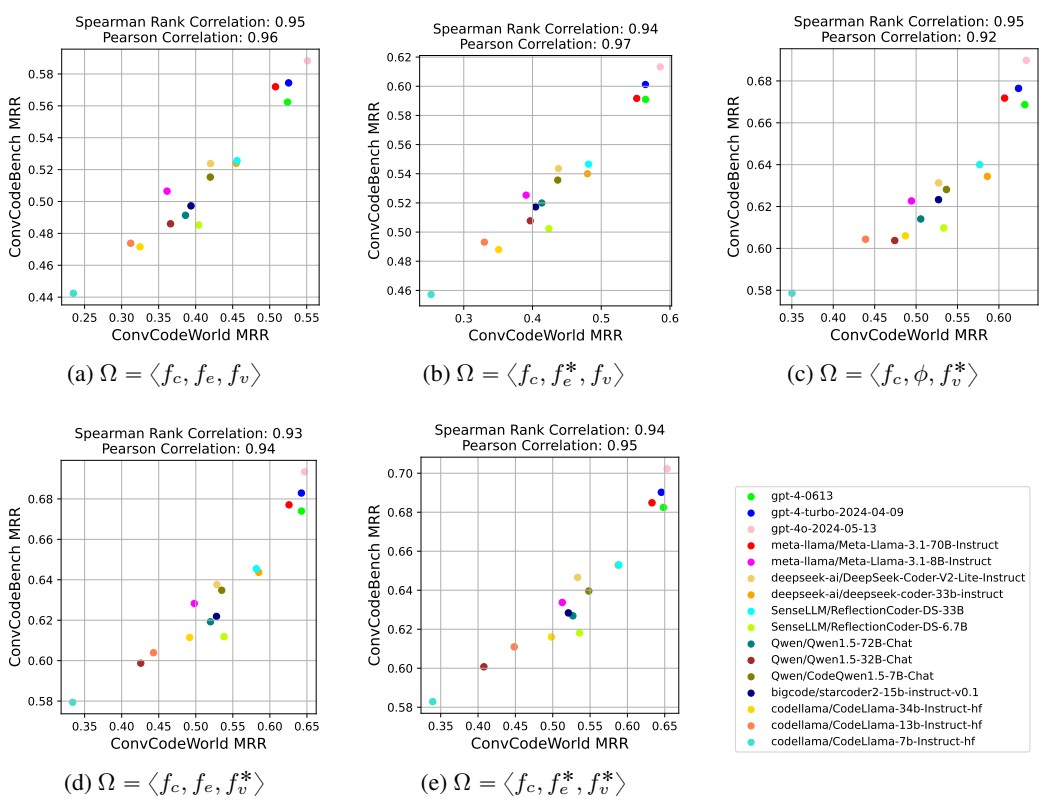

Figure 7: Correlation between MRR on CONVCODEBENCH (ref. DeepSeek-Coder-6.7B-Instruct) and MRR on CONVCODEWORLD with different feedback combinations $\Omega$.

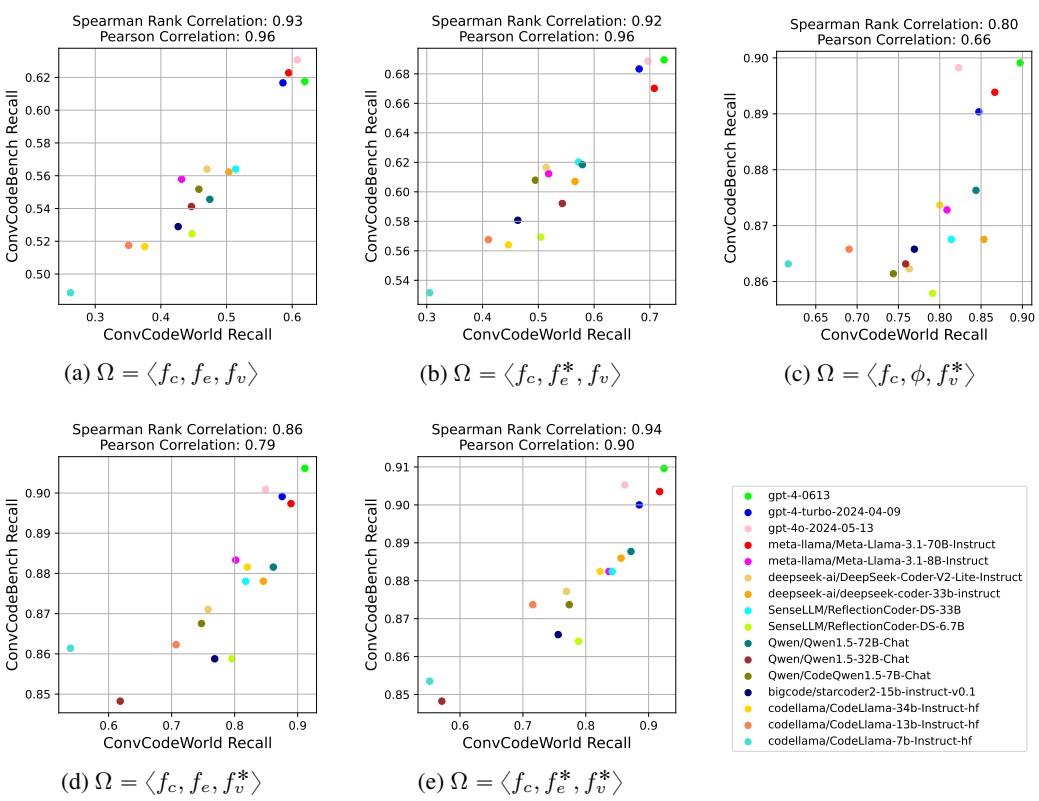

Figure 8: Correlation between Recall on CONVCODEBENCH (ref. DeepSeek-Coder-6.7B-Instruct) and Recall on CONVCODEWORLD with different feedback combinations $\Omega$.

### H.1.2 REFERENCE MODEL: GPT-4-0613

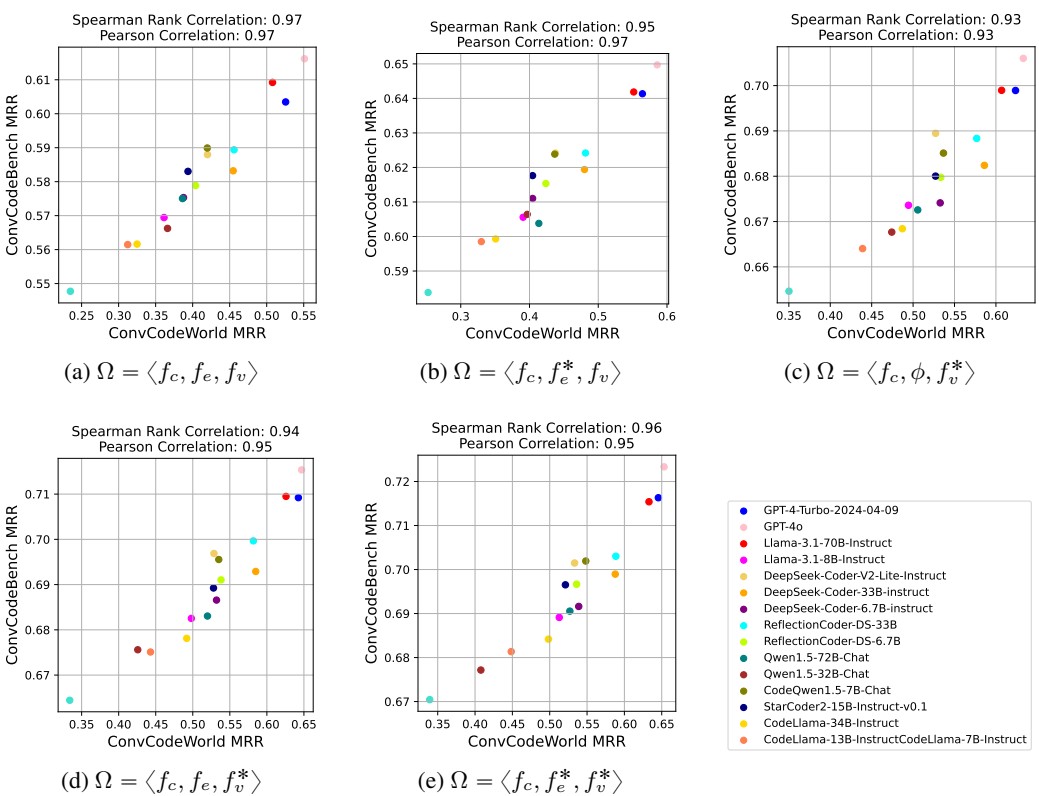

Figure 9: Correlation between MRR on CONVCODEBENCH (ref. GPT-4-0613) and MRR on CONVCODEWORLD with different feedback combinations $\Omega$.

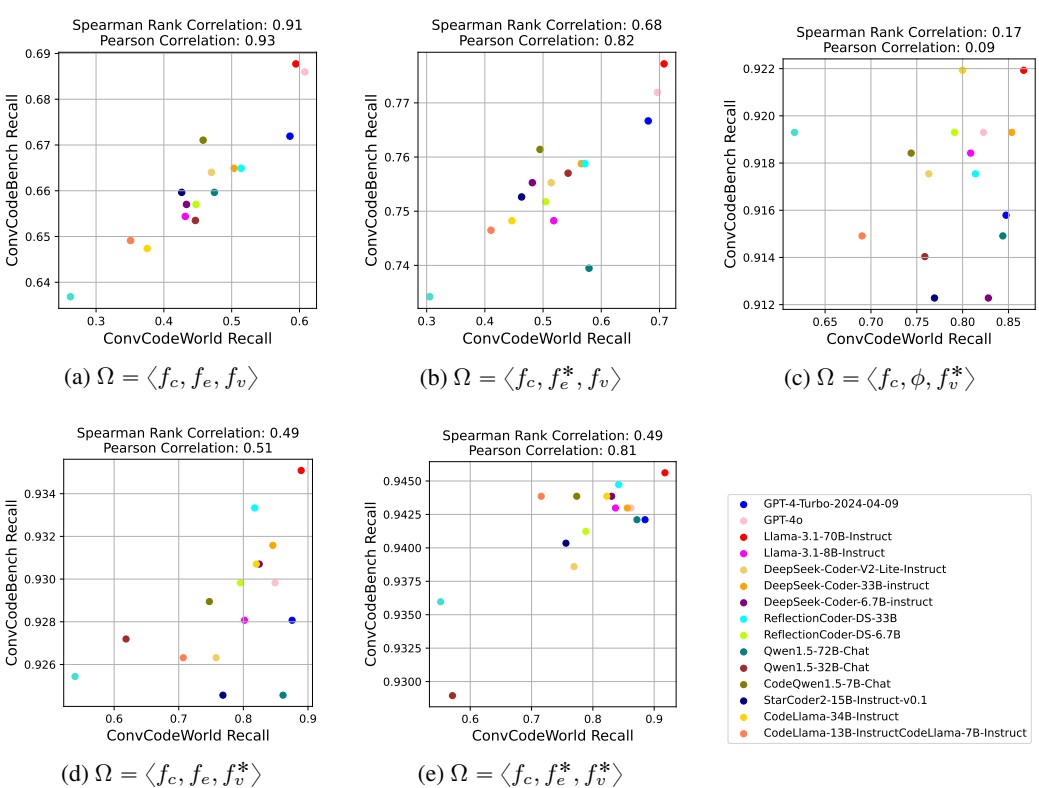

Figure 10: Correlation between Recall on CONVCODEBENCH (ref. GPT-4-0613) and Recall on CONVCODEWORLD with different feedback combinations $\Omega$.

# I  IN-CONTEXT LEARNING EXAMPLES

## I.1  NOVICE-LEVEL VERBAL FEEDBACK

```
Example Input:
Sort a list of integers in ascending order. The function should take a
↪   list of integers and return a sorted list. Ensure that the function
↪   handles negative numbers and zeros correctly. Check if the function's
↪   output is a sorted list.
```python
>>> sorted_list = sort_func([3, -1, 0, 5, -10, 2])
>>> sorted_list
[-10, -1, 0, 2, 3, 5]
```
You should write self-contained code starting with:
```python
def sort_func(int_list):
```

Ground Truth Code:
```python
def sort_func(int_list):
    return sorted(int_list)
```

Previous Code:
```python
def sort_func(int_list):
    for i in range(len(int_list)):
        for j in range(len(int_list) - 1):
            if int_list[j] < int_list[j + 1]:
                int_list[j], int_list[j + 1] = int_list[j + 1],
                ↪   int_list[j]
    return int_list

test_list = [3, -1, 0, 5, -10, 2]
print(sort_func(test_list))
```

Compilation Feedback:
Traceback (most recent call last):
  File "tmp.py", line 6
    return int_list
                   ^
IndentationError: unindent does not match any outer indentation level

Reasoning:
Let's think step by step in order to produce the novice-level
↪   `user_feedback`.

User Feedback:
I got this error:
```
Traceback (most recent call last):
  File "tmp.py", line 6
    return int_list
                   ^
IndentationError: unindent does not match any outer indentation level
```
I think there might be something wrong with the indentation of the
↪   `return int_list` line. Also, I think there exists a simpler way to
↪   sort a list in Python.
```

Figure 11: The in-context example of $f_v$ generation where $\Omega = \langle f_c, \phi, f_v \rangle$.

```
1   Example Input:
2   Sort a list of integers in ascending order. The function should take a
    ↪   list of integers and return a sorted list. Ensure that the function
    ↪   handles negative numbers and zeros correctly. Check if the function's
    ↪   output is a sorted list.
3   ```python
4   >>> sorted_list = sort_func([3, -1, 0, 5, -10, 2])
5   >>> sorted_list
6   [-10, -1, 0, 2, 3, 5]
7   ```
8   You should write self-contained code starting with:
9   ```python
10  def sort_func(int_list):
11  ```
12
13  Previous Code:
14  ```python
15  def sort_func(int_list):
16      for i in range(len(int_list)):
17          for j in range(len(int_list) - 1):
18              if int_list[j] < int_list[j + 1]:
19                  int_list[j], int_list[j + 1] = int_list[j + 1],
                    ↪   int_list[j]
20      return int_list
21
22  test_list = [3, -1, 0, 5, -10, 2]
23  print(sort_func(test_list))
24  ```
25
26  Compilation Feedback:
27  No syntax errors
28
29  Execution Feedback:
30  TEST_CASE_1
31  Traceback (most recent call last):
32    File "__test__.py", line 78, in test_case_1
33  AssertionError: sort_func([3, -1, 0, 5, -10, 2]) != [-10, -1, 0, 2, 3, 5]
34
35  Reasoning:
36  Let's think step by step in order to produce the novice-level
    ↪   `user_feedback`.
37
38  User Feedback:
39  I got this error:
40  ```
41  Traceback (most recent call last):
42    File "__test__.py", line 78, in test_case_1
43  AssertionError: sort_func([3, -1, 0, 5, -10, 2]) != [-10, -1, 0, 2, 3, 5]
44  ```
45  It seems like the function didn't sort the list correctly. The list is in
    ↪   descending order instead of ascending order. Also, I think there
    ↪   exists a simpler way to sort a list in Python.
```

Figure 12: The in-context example of $f_v$ generation where $\Omega = \langle f_c, f_e[*], f_v \rangle$.

## I.2 EXPERT-LEVEL VERBAL FEEDBACK

```
1   Example Input:
2   Sort a list of integers in ascending order. The function should take a list of integers and
    ↪  return a sorted list. Ensure that the function handles negative numbers and zeros
    ↪  correctly. Check if the function's output is a sorted list.
3   ```python
4   >>> sorted_list = sort_func([3, -1, 0, 5, -10, 2])
5   >>> sorted_list
6   [-10, -1, 0, 2, 3, 5]
7   ```
8   You should write self-contained code starting with:
9   ```python
10  def sort_func(int_list):
11  ```
12
13  Ground Truth Code:
14  ```python
15  def sort_func(int_list):
16      return sorted(int_list)
17  ```
18
19  Previous Code:
20  ```python
21  def sort_func(int_list):
22      for i in range(len(int_list)):
23          for j in range(len(int_list) - 1):
24              if int_list[j] < int_list[j + 1]:
25                  int_list[j], int_list[j + 1] = int_list[j + 1], int_list[j]
26      return int_list
27
28  test_list = [3, -1, 0, 5, -10, 2]
29  print(sort_func(test_list))
30  ```
31
32  Reasoning:
33  Let's think step by step in order to produce the user_feedback. We need to compare the
    ↪  `previous_code` with the `ground_truth_code` and identify specific areas where the
    ↪  `previous_code` deviates from the expected behavior. We should focus on the sorting
    ↪  algorithm, handling of edge cases like negative numbers and zeros, and the overall
    ↪  performance of the code.
34
35  User Feedback:
36  1. **Sorting Algorithm**: The `previous_code` implements a bubble sort algorithm, but it sorts
    ↪  the list in descending order due to the condition `if int_list[j] < int_list[j + 1]`. The
    ↪  correct approach should use `if int_list[j] > int_list[j + 1]` to sort in ascending order.
37
38  2. **Edge Case Handling**: While the `previous_code` attempts to handle all integers, it
    ↪  incorrectly sorts the list in the reverse order. Ensure that the sorting condition
    ↪  correctly compares elements to achieve ascending order.
39
40  3. **Code Simplicity and Readability**: The bubble sort implementation in the `previous_code`
    ↪  is more complex and less readable compared to using the built-in `sorted()` function.
    ↪  Simplifying the code to use `sorted()` will enhance readability and maintainability.
41
42  4. **Performance**: The bubble sort algorithm used in the `previous_code` has a time
    ↪  complexity of O(n^2), which can be inefficient for large lists. The `sorted()` function,
    ↪  on the other hand, uses Timsort, which has a time complexity of O(n log n) and is more
    ↪  suitable for sorting large datasets.
43
44  5. **Test Case Verification**: The `previous_code` includes a print statement to verify the
    ↪  function's output. While this is useful for debugging, consider using assertions or test
    ↪  functions to automate the verification process in a more scalable way.
45
46  Overall, the `previous_code` functions incorrectly by sorting the list in descending order.
    ↪  Using the correct comparison in the bubble sort condition or utilizing Python's built-in
    ↪  sorting capabilities would improve the function's correctness and performance.
47
```

Figure 13: The in-context example of $f_v^*$ generation where $\Omega = \langle f_c, \phi, f_v^* \rangle$.

```
1   Example Input:
2   Sort a list of integers in ascending order. The function should take a list of integers and
    ↪   return a sorted list. Ensure that the function handles negative numbers and zeros
    ↪   correctly. Check if the function's output is a sorted list.
3   ```python
4   >>> sorted_list = sort_func([3, -1, 0, 5, -10, 2])
5   >>> sorted_list
6   [-10, -1, 0, 2, 3, 5]
7   ```
8   You should write self-contained code starting with:
9   ```python
10  def sort_func(int_list):
11  ```
12
13  Ground Truth Code:
14  ```python
15  def sort_func(int_list):
16      return sorted(int_list)
17  ```
18
19  Previous Code:
20  ```python
21  def sort_func(int_list):
22      for i in range(len(int_list)):
23          for j in range(len(int_list) - 1):
24              if int_list[j] < int_list[j + 1]:
25                  int_list[j], int_list[j + 1] = int_list[j + 1], int_list[j]
26      return int_list
27
28  test_list = [3, -1, 0, 5, -10, 2]
29  print(sort_func(test_list))
30  ```
31
32  Execution Feedback:
33  TEST_CASE_1
34  Traceback (most recent call last):
35    File "__test__.py", line 78, in test_case_1
36  AssertionError: sort_func([3, -1, 0, 5, -10, 2]) != [-10, -1, 0, 2, 3, 5]
37
38  Reasoning:
39  Let's think step by step in order to produce the user_feedback. We need to compare the
    ↪   `previous_code` with the `ground_truth_code` and identify specific areas where the
    ↪   `previous_code` deviates from the expected behavior. We should focus on the sorting
    ↪   algorithm, handling of edge cases like negative numbers and zeros, and the overall
    ↪   performance of the code.
40
41  User Feedback:
42  1. **Sorting Algorithm**: The `previous_code` implements a bubble sort algorithm, but it sorts
    ↪   the list in descending order due to the condition `if int_list[j] < int_list[j + 1]`. The
    ↪   correct approach should use `if int_list[j] > int_list[j + 1]` to sort in ascending order.
43
44  2. **Edge Case Handling**: While the `previous_code` attempts to handle all integers, it
    ↪   incorrectly sorts the list in the reverse order. Ensure that the sorting condition
    ↪   correctly compares elements to achieve ascending order.
45
46  3. **Code Simplicity and Readability**: The bubble sort implementation in the `previous_code`
    ↪   is more complex and less readable compared to using the built-in `sorted()` function.
    ↪   Simplifying the code to use `sorted()` will enhance readability and maintainability.
47
48  4. **Performance**: The bubble sort algorithm used in the `previous_code` has a time
    ↪   complexity of O(n^2), which can be inefficient for large lists. The `sorted()` function,
    ↪   on the other hand, uses Timsort, which has a time complexity of O(n log n) and is more
    ↪   suitable for sorting large datasets.
49
50  5. **Test Case Verification**: The `previous_code` includes a print statement to verify the
    ↪   function's output. While this is useful for debugging, consider using assertions or test
    ↪   functions to automate the verification process in a more scalable way.
51
52  Overall, the `previous_code` functions incorrectly by sorting the list in descending order.
    ↪   Using the correct comparison in the bubble sort condition or utilizing Python's built-in
    ↪   sorting capabilities would improve the function's correctness and performance.
53
```

Figure 14: The in-context example of $f_v^*$ generation where $\Omega = \langle f_c, f_e[*], f_v^* \rangle$.

```
1  You are given input, previous_code, execution_feedback to simulate user feedback that compares
   ↪  the `previous_code` and the `ground_truth_code`.
2  Your task is to provide the simulated `user_feedback` that highlights specific areas where the
   ↪  `previous_code` deviates from the `ground_truth_code` and suggests improvements or
   ↪  corrections.
3  - You SHOULD NOT leak `ground_truth_code` in the simulated user feedback.
4  - Do not generate updated code.
5  - Do not reveal that you can access the `ground_truth_code`. Only indirect information is
   ↪  allowed.
```

Figure 15: Prompt used for expert feedback generation in the feedback combination $\Omega = \langle f_c, [f_e|f_e^*], f_v^* \rangle$.

## J    CASE STUDY

**Input Code**

```python
import smtplib
from email.message import EmailMessage
import getpass
import socket

SERVER_ADDRESS = "localhost"
SERVER_PORT = 25
BUFFER_SIZE = 1024
SMTP_SERVER = "smtp.gmail.com"
SMTP_PORT = 587

def task_func(client_socket):
    # Ensure the socket is connected
    if not client_socket:
        print("Client socket is not connected.")
        return

    # Receive the message from the client
    try:
        message = client_socket.recv(BUFFER_SIZE).decode()
    except socket.error as e:
        print(f"Failed to receive message from client: {e}")
        return

    # If the message is empty, do not send the email
    if not message:
        print("Received an empty message. Not sending email.")
        return

    # Ask for the sender's email, recipient's email, and sender's email password for
        authentication
    sender_email = input("Enter sender's email: ")
    recipient_email = input("Enter recipient's email: ")
    password = getpass.getpass("Enter sender's email password: ")

    # Create the email message
    email_message = EmailMessage()
    email_message.set_content(message)
    email_message["Subject"] = "Message from client"
    email_message["From"] = sender_email
    email_message["To"] = recipient_email

    # Connect to the SMTP server and send the email
    try:
        with smtplib.SMTP(SMTP_SERVER, SMTP_PORT) as smtp:
            smtp.starttls()
            smtp.login(sender_email, password)
            smtp.send_message(email_message)
            print("Email sent successfully.")
    except Exception as e:
        print(f"Failed to send email: {e}")
```

**Execution Feedback with Partial Test Coverage ($f_e$)**

```
TEST_SMTP_SERVER_CONNECTION_ERROR
Traceback (most recent call last):
  File "/notebooks/miniconda/envs/bigcodebench/lib/python3.9/unittest/mock.py", line 1336, in patched
    return func(*newargs, **newkeywargs)
  File "__test__.py", line 125, in test_smtp_server_connection_error
  File "__test__.py", line 39, in task_func
  File "/tmp/utils.py", line 251, in readline
    raise IOError
OSError

TEST_SUCCESSFUL_EMAIL_SEND
Traceback (most recent call last):
  File "/notebooks/miniconda/envs/bigcodebench/lib/python3.9/unittest/mock.py", line 1336, in patched
    return func(*newargs, **newkeywargs)
  File "__test__.py", line 81, in test_successful_email_send
  File "__test__.py", line 39, in task_func
  File "/tmp/utils.py", line 251, in readline
    raise IOError
OSError
```

Figure 16: Case study of GPT-4 on CONVCODEWORLD ($\Omega = \langle f_c, f_e, f_v \rangle$; $t = 7$): Input Code and $f_e$.

## Novice-Level Verbal Feedback ($f_v$)

```
1   I got this error:
2   ```
3   Traceback (most recent call last):
4     File "/notebooks/miniconda/envs/bigcodebench/lib/python3.9/unittest/mock.py", line 1336, in patched
5       return func(*newargs, **newkeywargs)
6     File "__test__.py", line 125, in test_smtp_server_connection_error
7     File "__test__.py", line 39, in task_func
8     File "/tmp/utils.py", line 251, in readline
9       raise IOError
10  OSError
11
12  Traceback (most recent call last):
13    File "/notebooks/miniconda/envs/bigcodebench/lib/python3.9/unittest/mock.py", line 1336, in patched
14      return func(*newargs, **newkeywargs)
15    File "__test__.py", line 81, in test_successful_email_send
16    File "__test__.py", line 39, in task_func
17    File "/tmp/utils.py", line 251, in readline
18      raise IOError
19  OSError
20  ```
21  It seems like there is an issue with the socket connection or the way the code is handling the socket. The
    ↪  `OSError` exceptions are being raised during the execution of the `task_func` function.
```

## Refined Code by Feedback

```python
1   import smtplib
2   from email.message import EmailMessage
3   import getpass
4   import socket
5
6   SERVER_ADDRESS = "localhost"
7   SERVER_PORT = 25
8   BUFFER_SIZE = 1024
9   SMTP_SERVER = "smtp.gmail.com"
10  SMTP_PORT = 587
11
12  def task_func(client_socket):
13      # Ensure the socket is connected
14      if not client_socket:
15          print("Client socket is not connected.")
16          return
17
18      # Receive the message from the client
19      try:
20          message = client_socket.recv(BUFFER_SIZE).decode()
21      except socket.error as e:
22          print(f"Failed to receive message from client: {e}")
23          return
24
25      # If the message is empty, do not send the email
26      if not message:
27          print("Received an empty message. Not sending email.")
28          return
29
30      # Ask for the sender's email, recipient's email, and sender's email password for
              authentication
31      sender_email = input("Enter sender's email: ")
32      recipient_email = input("Enter recipient's email: ")
33      password = getpass.getpass("Enter sender's email password: ")
34
35      # Create the email message
36      email_message = EmailMessage()
37      email_message.set_content(message)
38      email_message["Subject"] = "Message from client"
39      email_message["From"] = sender_email
40      email_message["To"] = recipient_email
41
42      # Connect to the SMTP server and send the email
43      try:
44          with smtplib.SMTP(SMTP_SERVER, SMTP_PORT) as smtp:
45              smtp.starttls()
46              smtp.login(sender_email, password)
47              smtp.send_message(email_message)
48              print("Email sent successfully.")
49      except Exception as e:
50          print(f"Failed to send email: {e}")
```

Figure 17: Case study of GPT-4-0613 on CONVCODEWORLD ($\Omega = \langle f_c, f_e, f_v \rangle$; turn 7): $f_v$ and Refined Code.

**Input Code**

```
1  import matplotlib.pyplot as plt
2  import numpy as np
3  def task_func(n, seed=0):
4      np.random.seed(seed)
5      points = [(x, y) for x, y in np.random.rand(n, 2)]
6
7      fig, ax = plt.subplots()
8      ax.scatter([x for x, y in points], [y for x, y in points])
9      ax.set_title("Scatter_plot_of_random_points")
10     ax.set_xlabel("X")
11     ax.set_ylabel("Y")
12
13     return fig, points
```

**Expert-Level Verbal Feedback ($f_v^*$)**

```
1  1. **Random Points Generation**: The `previous_code` generates random points using a list
   ↪   comprehension with `np.random.rand(n, 2)`, which is a valid approach. However, it can be
   ↪   simplified by directly using `np.random.rand(n)` for both x and y coordinates separately.
   ↪   This makes the code more readable and aligns with common practices.

2
3  2. **Plotting Process**: The `previous_code` correctly plots the scatter plot using
   ↪   `ax.scatter()`. However, it extracts x and y coordinates from the `points` list using list
   ↪   comprehensions. This step can be avoided by directly using the arrays generated for x and
   ↪   y coordinates, which simplifies the code.

4
5  3. **Return Values**: The `previous_code` correctly returns the figure and the list of points.
   ↪   However, the generation of points can be streamlined by using `zip(x, y)` directly on the
   ↪   generated x and y arrays, which is more efficient and concise.

6
7  4. **Code Simplicity and Readability**: The `previous_code` is functional but can be
   ↪   simplified for better readability. Using direct array operations for x and y coordinates
   ↪   and avoiding unnecessary list comprehensions will make the code cleaner and easier to
   ↪   understand.

8
9  5. **Reproducibility**: Both the `previous_code` and the expected implementation correctly set
   ↪   the random seed for reproducibility, which is good practice.

10
11 Overall, while the `previous_code` is functional, it can be improved for simplicity and
   ↪   readability by directly using the generated x and y arrays for plotting and point
   ↪   generation. This will make the code more concise and maintainable.
```

**Refined Code by Feedback**

```
1  import matplotlib.pyplot as plt
2  import numpy as np
3  def task_func(n, seed=0):
4      np.random.seed(seed)
5      x, y = np.random.rand(2, n)
6      points = list(zip(x, y))
7
8      fig, ax = plt.subplots()
9      ax.scatter(x, y)
10     ax.set_title("Scatter_plot_of_random_points")
11     ax.set_xlabel("X")
12     ax.set_ylabel("Y")
13
14     return fig, points
```

Figure 18: Case study of DeepSeek-Coder-6.7B-Instruct on CONVCODEWORLD ($\Omega = \langle f_c, \phi, f_v^* \rangle$; turn 1).

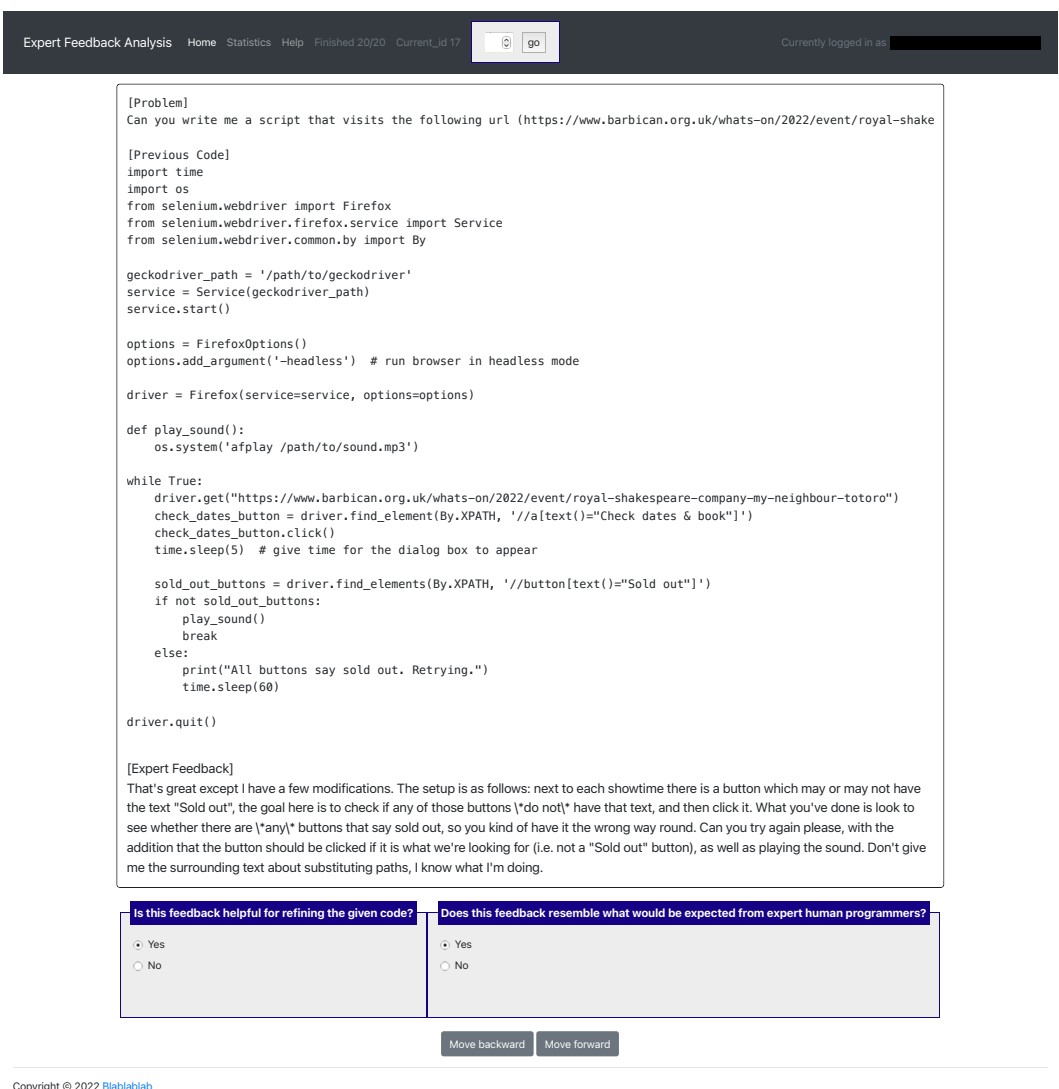

Figure 19: Annotation platform (Pei et al., 2022) for collecting human evaluation of expert verbal feedback.

```
1  1. **Configuration File Reading**: The `previous_code` correctly reads
   ↪  the configuration file using `configparser`. However, ensure that the
   ↪  configuration file path is valid and exists before attempting to read
   ↪  it. This is not explicitly checked in the `previous_code`.
2
3  2. **Directory Existence Check**: The `previous_code` uses
   ↪  `os.path.exists(project_dir)` to check if the project directory
   ↪  exists. While this works, it is more appropriate to use
   ↪  `os.path.isdir(project_dir)` to specifically check for directory
   ↪  existence, as it is more semantically correct.
4
5  3. **ZIP Archive Creation**: The `previous_code` attempts to create the
   ↪  ZIP archive using `shutil.make_archive(project_dir, 'zip',
   ↪  archive_dir)`. This is incorrect because `shutil.make_archive`
   ↪  expects the base name of the archive and the root directory to
   ↪  archive. The correct usage should be
   ↪  `shutil.make_archive(base_name=os.path.splitext(zip_file_path)[0],
   ↪  format='zip', root_dir=project_dir)`.
6
7  4. **Exception Handling**: The `previous_code` raises a generic
   ↪  `Exception` if the ZIP archive creation fails. While this is
   ↪  acceptable, it is better to provide a more specific error message
   ↪  indicating the failure reason. Additionally, ensure that the ZIP file
   ↪  is actually created by checking its existence after the
   ↪  `shutil.make_archive` call.
8
9  5. **Return Value**: The `previous_code` correctly returns `True` if the
   ↪  ZIP archive is successfully created. However, it should also ensure
   ↪  that the ZIP file exists before returning `True`.
10
11 6. **Code Simplicity and Readability**: The `previous_code` includes a
   ↪  detailed docstring, which is good practice. However, the actual
   ↪  implementation can be simplified and made more readable by following
   ↪  the correct usage of `shutil.make_archive` and ensuring proper
   ↪  exception handling.
12
13 Overall, the `previous_code` has the right structure but needs
   ↪  corrections in the directory existence check, ZIP archive creation,
   ↪  and exception handling to function correctly.
```

Figure 20: An example case that the feedback guides toward the ground truth without explicitly referencing it.

```
1  1. **Class Name**: The class name in the `previous_code` is
   ↪  `EmailHandler`, but it should be `EmailRequestHandler` to match the
   ↪  `ground_truth_code`.

2
3  2. **Content-Type Check**: Instead of directly checking the
   ↪  `Content-Type` header, use `cgi.parse_header` to parse the header and
   ↪  then check if `ctype != 'application/json'`.

4
5  3. **Error Handling for Content-Type**: When the `Content-Type` is not
   ↪  `application/json`, simply send a 400 response and end headers
   ↪  without writing a message to the response body.

6
7  4. **Reading Content-Length**: Use `length =
   ↪  int(self.headers.get('content-length'))` instead of `content_length =
   ↪  int(self.headers.get('Content-Length', 0))`.

8
9  5. **JSON Decoding**: When catching `json.JSONDecodeError`, send a 400
   ↪  response and end headers without writing a message to the response
   ↪  body.

10
11 6. **Missing Fields Check**: When required fields are missing, send a 400
   ↪  response and end headers without writing a message to the response
   ↪  body.

12
13 7. **SMTP Authentication Error Handling**: When catching
   ↪  `smtplib.SMTPAuthenticationError`, send a 535 response and end
   ↪  headers without writing a message to the response body.

14
15 8. **General Exception Handling**: Remove the general exception handler
   ↪  that sends a 500 response, as it is not present in the
   ↪  `ground_truth_code`.

16
17 By making these changes, the `previous_code` will align more closely with
   ↪  the `ground_truth_code`.
```

Figure 21: An example case that the feedback directly references ground truth, leading to "leakage".

