# OpenReview forum: "ConvCodeWorld: Benchmarking Conversational Code Generation in Reproducible Feedback Environments"
_ICLR.cc/2025/Conference — ICLR 2025 Poster_

### Official Review · Reviewer_acNZ · 2024-10-22

**Soundness:** 3
**Presentation:** 4
**Contribution:** 3
**Rating:** 6
**Confidence:** 4

**Summary:**

This work presents ConvCodeWorld, a benchmark for evaluating how well language models use feedback to address a code generation problem across multiple turns. While multi-turn interactive code generation work exists, the impact of the type of feedback on model performance is not as thoroughly examined. To carry out this study, the authors formulate three types of feedback (compilation, execution, verbal) and construct a pipeline (ConvCodeBench) that bootstraps existing code generation benchmarks to construct a static benchmark that follows the ConvCodeWorld formulation. In the results, the authors find that generally, closed source models perform best. Through ablations of different feedback techniques and open source models, the authors also discuss trends around the topics of generalization, the impact of turns, and the correlation between ConvCodeBench and ConvCodeWorld.

**Strengths:**

- Feedback in code generation feels like a fairly understudied area. The premise of the paper is quite interesting in my opinion. The results are also neat, particularly how different models perform well under different feedback scenarios.
- The fact that ConvCodeBench bootstraps existing code generation benchmarks is quite compelling. While the authors apply this towards BigCodeBench, it sounds like it would work for a wide variety of similar benchmarks.
- The evaluation is quite comprehensive, and the experimental setup is easy to understand + sound.
- Section 4.2.3 - Given the knowledge about how ReflectionCoder was trained, it is quite interesting to see that the model’s performance varies with the type of feedback provided.
- The authors’ experiments reveal a number of interesting insights. I particularly liked how ConvCodeBench performance is shown to be a good proxy for ConvCodeWorld performance, and the relationship between MRR and recall.

**Weaknesses:**

- Section 2.1.1 is written in a way that a lot of references are made to the appendix, which requires a lot of jumping back and forth. As Appendix A.3 is referenced multiple times, it might be worth considering just putting the information justifying the claims discussed here directly in the main paper.
- The mathematical notation in the paper is formalized correctly. From a readability standpoint, I would have preferred to just read what feedback types correspond to what results, as opposed to having to map symbols back to their corresponding feedback. Section 4.2.1 was a bit difficult to parse because of this. Of course, this is just my opinion.
- For Takeaway 2 (Impact of Expert-level…) in Section 4.2.1 - It was helpful to be told these observations as opposed to reading them from the table, but an explanation for why such trends were observed feels a bit lacking (“struggle to utilize complex feedback” feels a bit coarse-grained as a justification for a lot of observations)
- I’m not sure the takeaway – that feedback helps weaker models perform stronger than stronger models in a zero-shot setting – is that surprising given prior work (e.g. Mint, InterCode discuss this to some capacity).

**Questions:**

- How are you detecting whether ground truth code is leaked by the GPT 4o based expert? From reading A.3, I couldn’t quite understand this. Is the `ground_truth_code` function doing an exact match lookup in GPT 4o’s feedback? Or is there another implementation? Table 7 seems to be justifying that the verbal feedback doesn’t leak ground truth code because of downstream task performance differences, instead of a direct examination of the verbal feedback’s content.
- What does Table 9 mean? When it is first referenced in Section 2.1.1, the phi character does not appear to be defined at that point in the paper. Also, GPT 4/4o models are tested exclusively. Any reason why Claude / Llama / other models were not used?
- I agree with the justification that human annotators are more costly than GPT 4o. However, is it possible that novice/expert-level feedback looks very different between humans and language models? I think a justification for why LMs are a good proxy for human feedback for ConvCodeWorld would be good assurance.
- Line 247: Which two settings are being referenced for this comparison?

---

> ### Author Response · Authors · 2024-11-22
> **Author Response to Reviewer acNZ (Q1, Q2)**
>
> We sincerely thank you for your detailed review.
> > Q1-1. How are you detecting whether ground truth code is leaked by the GPT 4o based expert? From reading A.3, I couldn’t quite understand this. Is the ground_truth_code function doing an exact match lookup in GPT 4o’s feedback? Or is there another implementation?
>
> > Q1-2. Table 7 seems to be justifying that the verbal feedback doesn’t leak ground truth code because of downstream task performance differences, instead of a direct examination of the verbal feedback’s content.
>
> **A1-1: Ground truth code leakage detection is done by canary sequence**:
> - We considered as leakage if the feedback simulator includes a **canary sequence** (while it is misinterpreted as a function in the review). This use of canary sequences—``ground_truth_code`` given in the prompt (see Example 1)—is common in testing for training data or prompt leakage in LLMs [1, 2, 3, 4, 5] (see response towards Reviewer cHoy for a leakage example).
> - **How ConvCodeWorld minimizes the leakage?**: We chose GPT-4o for the simulator since its significantly lower rate of ground truth code mention (2.5%) compared to GPT-4-0613 (51.1%) and GPT-4-Turbo-2024-04-09 (31.4%), and only 0.1% of refined code inclusion (see Table 8).
>
> We will clarify this behavior in Appendix A.3 and provide examples to further illustrate these observations.
>
> **A1-2: Table 8 is a direct examination**
> - **Table 8** provides a **direct examination** of the verbal feedback's content. In Table 8, we report the instances of leakage as defined by the occurrence of `ground_truth_code` in the feedback.
> - From the direct analysis the content of the verbal feedback for specific terms, we selected GPT-4o-2024-05-13 as the expert feedback simulator, which has **significantly lower ground truth code leakage** compared to other models.
>
> Example 1. Prompt used for expert feedback generation in the feedback combination $\langle f_c, [f_e|f_e^*], f_v^* \rangle$.
> ```
> You are given input, previous_code, execution_feedback to simulate user feedback that compares the `previous_code` and the `ground_truth_code`.
> Your task is to provide the simulated `user_feedback` that highlights specific areas where the `previous_code` deviates from the `ground_truth_code` and suggests improvements or corrections.
> - You SHOULD NOT leak `ground_truth_code` in the simulated user feedback.
> - Do not generate updated code.
> - Do not reveal that you can access the `ground_truth_code`. Only indirect information is allowed.
> ```
>
> > Q2-1. What does Table 9 mean? When it is first referenced in Section 2.1.1, the phi character does not appear to be defined at that point in the paper.
>
> > Q2-2. Also, GPT 4/4o models are tested exclusively. Any reason why Claude / Llama / other models were not used?
>
> **A2-1: Clarification on Table 9:**
> - Table 9 evaluates different models as potential verbal feedback simulators. In this table:
>     - Each **row** represents a model used to provide verbal feedback.
>     - Each **column** represents a model that utilizes this feedback to refine code.
> - By comparing the performance across columns, we assess the effectiveness of the feedback provided by each simulator.
> - Regarding the symbol $\phi$, we will provide a clear explanation in the caption of Table 9 in the next version.
>
> **A2-2: Why GPT models?**:
> - Previous findings in studies like MINT (see Table 4 in the MINT paper) demonstrated that **model performance is crucial for effective feedback**.
> - During the setup of our experimental settings, there was no such powerful open-weight models (see response towards Reviewer cHoy for using open-source models as verbal feedback simulators).
> - Based on that, we decided to consider high-performing closed-source models, such as GPT-4 and GPT-4o, which allowed us to achieve reliable results.
> - Claude models were not considered due to resource constraints and computational credits supported by our organization.

---

> > ### Author Response · Authors · 2024-11-22
> > **Author Response to Reviewer acNZ (Q3, Q4, W1, W2, W3)**
> >
> > > Q3. I agree with the justification that human annotators are more costly than GPT 4o. However, is it possible that novice/expert-level feedback looks very different between humans and language models? I think a justification for why LMs are a good proxy for human feedback for ConvCodeWorld would be good assurance.
> >
> > **A3: We provide the following justifications**
> >
> > - **Expert Feedback**: As mentioned in lines 655-657, our approach to generating expert feedback aligns with MINT's "natural language" feedback, which incorporates ground truth code information in feedback generation. MINT's human evaluation results (see Table 5 in the MINT paper) indicate that this type of feedback is both **helpful and perceived as human-like**, supporting the validity of our approach.
> >
> > - **Novice Feedback**: As shown in Figure 16, novice feedback primarily consists of **verbalized explanations** of other feedback types (e.g., execution feedback), which are reasonablely expected from novice programmers.
> >
> > While a large-scale HCI study involving real experts was beyond our resource constraints, we aim to support such future analyses by **open-sourcing both ConvCodeWorld and ConvCodeBench**. ConvCodeBench includes all generated novice and expert feedback, enabling researchers to compare simulated feedback with human feedback in subsequent studies.
> >
> >
> > > Q4. Line 247: Which two settings are being referenced for this comparison?
> >
> > The two settings being compared are ConvCodeBench (static) and ConvCodeWorld (live), as illustrated in Figures 2 and 3. We will clarify this in the revised manuscript.
> >
> >
> > > W1. Section 2.1.1 is written in a way that a lot of references are made to the appendix, which requires a lot of jumping back and forth. As Appendix A.3 is referenced multiple times, it might be worth considering just putting the information justifying the claims discussed here directly in the main paper.
> >
> > > W2. The mathematical notation in the paper is formalized correctly. From a readability standpoint, I would have preferred to just read what feedback types correspond to what results, as opposed to having to map symbols back to their corresponding feedback. Section 4.2.1 was a bit difficult to parse because of this. Of course, this is just my opinion.
> >
> > Thanks for the suggestion. We will move Appendix A.3 directly into the main context, and will improve the readability in the next version.
> >
> >
> > > W3. For Takeaway 2 (Impact of Expert-level…) in Section 4.2.1 - It was helpful to be told these observations as opposed to reading them from the table, but an explanation for why such trends were observed feels a bit lacking (“struggle to utilize complex feedback” feels a bit coarse-grained as a justification for a lot of observations)
> >
> > **Two possible reasons for the observed "struggle to utilize feedback"**
> >
> > Though we could not fully present rationales due to space limitation, here are two possible reasons for models when they struggle to utilize complex feedback:
> >
> > 1. **Limited Model size**:
> >     - Smaller models, such as ReflectionCoder-DS-6.7B, may lack the capacity to process and integrate complex information effectively, which could limit performance improvements even when execution feedback is included (35.2 $\rightarrow$ 37.7).
> >     - In contrast, their bigger versions like ReflectionCoder-DS-33B demonstrated performance gains with execution feedback (41.6 $\rightarrow$ 45.3).
> >     - Mixed feedback types may distract small models further. When comparing Expert feedback only vs. Expert feedback + execution feedback. for Qwen1.5-Chat, the 72B model's performance improved with execution feedback, while the 32B model's performance deteriorated, which suggests that smaller models might become distracted when faced with multiple feedback signals simultaneously [6]. However, this distraction may be mitigated with well-designed training data, as even smaller models like Llama-3.1-8B-Instruct show improvements when provided with more execution feedback.
> >
> >
> > 2. **Limited Generalization Training**:
> >     - ReflectionCoder models were trained on a specific feedback combination, $\langle f_c, f_e^*, f_v \rangle$, limiting their adaptability to other feedback types (Section 4.2.3).
> >     - For example, with expert feedback, ReflectionCoder-DS-33B scores lower (81.4) than its base model DeepSeekCoder-33B-Instruct (85.4).
> >
> > We will include these explanations in the next version.

---

> > > ### Author Response · Authors · 2024-11-22
> > > **Author Response to Reviewer acNZ (W4)**
> > >
> > > > W4. I’m not sure the takeaway – that feedback helps weaker models perform stronger than stronger models in a zero-shot setting – is that surprising given prior work (e.g. Mint, InterCode discuss this to some capacity).
> > >
> > >
> > > Surprisingly, the support for the  takeaway you mentioned  is not fully substantiated by MINT or InterCode. (see our response to Reviewer 3Fw3).  This motivates our study to better understand when feedback is beneficial and when it is not, as the two examples contrast below:
> > > - Llama-3.1-8B-Instruct improves from 31.4 to 51.8 Recall with feedback ($\langle f_c, f_e^*, f_v \rangle$), **surpassing** GPT-4o-2024-05-13's no-feedback score (50.8).
> > > - DeepSeek-Coder-6.7B-Instruct, despite higher baseline performance (35.2 vs 31.4), reaches only 48.2 with identical feedback, **failing to surpass** GPT-4o-2024-05-13.
> > >
> > >
> > >
> > >
> > > [1] Reid, Machel, et al. "Gemini 1.5: Unlocking multimodal understanding across millions of tokens of context." arXiv preprint arXiv:2403.05530 (2024).
> > >
> > > [2] Achiam, Josh, et al. "Gpt-4 technical report." arXiv preprint arXiv:2303.08774 (2023).
> > >
> > > [3] Greshake, Kai, et al. "Not what you've signed up for: Compromising real-world llm-integrated applications with indirect prompt injection." Proceedings of the 16th ACM Workshop on Artificial Intelligence and Security. 2023.
> > >
> > > [4] Fabio Perez, & Ian Ribeiro (2022). Ignore Previous Prompt: Attack Techniques For Language Models. In NeurIPS ML Safety Workshop.
> > >
> > > [5] Divyansh Agarwal, Alexander Fabbri, Ben Risher, Philippe Laban, Shafiq Joty, and Chien-Sheng Wu. 2024. Prompt Leakage effect and mitigation strategies for multi-turn LLM Applications. In Proceedings of the 2024 Conference on Empirical Methods in Natural Language Processing: Industry Track, pages 1255–1275, Miami, Florida, US. Association for Computational Linguistics.
> > >
> > > [6] Nelson F. Liu, Kevin Lin, John Hewitt, Ashwin Paranjape, Michele Bevilacqua, Fabio Petroni, and Percy Liang. 2024. Lost in the Middle: How Language Models Use Long Contexts. Transactions of the Association for Computational Linguistics, 12:157–173.

---

> > > > ### Comment · Reviewer_acNZ · 2024-12-02
> > > > **Response to Authors**
> > > >
> > > > Thanks to the authors for the thorough responses to my questions. I appreciate the rebuttal and will maintain my score.

---

### Official Review · Reviewer_gBAk · 2024-10-23

**Soundness:** 3
**Presentation:** 4
**Contribution:** 3
**Rating:** 8
**Confidence:** 5

**Summary:**

The authors propose ConvCodeWorld, an environment for evaluating multi-turn code generation across 9 diverse feedback combinations. This entails compilation feedback, execution feedback (partial and full test coverage), and LLM-simulated verbal feedback (novice-level and expert-level). For this, they transform BigCodeBench-Instruct, a single-turn Python code generation benchmark. Additionally, they also propose a static benchmark to serve more practical purposes, ConvCodeBench, with pre-computed logs based on a weak model (CodeLlama-7B-Instruct). They find that performance on ConvCodeBench strongly correlates with performance on ConvCodeWorld. Using ConvCodeWorld, they evaluate 17 (3 closed-source and 14 open-source) LLMs of varying sizes. Through this, they highlight many key findings, including: (1)  Performance varies across feedback settings, often affecting the rankings, highlighting the need to select models based on the feedback available, (2) Weaker models with feedback can surpass single-turn SOTA models, (3) Models trained on a specific type of feedback struggle to generalize to unseen combinations of feedback, and (4) There is a tradeoff between an LLM's ability to solve problems in fewer turns (higher MRR) and solve many problems in total (high recall).

**Strengths:**

- As the authors have highlighted, evaluating multi-turn code generation is very difficult. By providing an environment with access to many diverse combinations of feedback, ConvCodeWorld could potentially be very useful to the research community, to hill-climb on multi-turn coding capabilities of LLMs.
- The authors have been very thorough about exploring different types of scenarios, with partial/full execution feedback, and novice/expert-level verbal feedback. This allows more fine-grained evaluation.
- There is an extensive analysis of performance of many different LLMs, and the key findings will likely be very interesting to the research community.

**Weaknesses:**

- While I agree that having a static benchmark like ConvCodeBench is useful, I am not convinced that it can be interpreted as a multi-turn evaluation set. Namely, my understanding is that it does not evaluate whether a model can iteratively improve its own prediction. The output from the previous turn is pre-defined, and so this seems to simply be trying to repair a program (which the model under test did not generate). If it fails to do it at a given turn, the output of that turn is ignored, and in the next turn, a fixed program is provided, which could be very different from the program the model under test generated in the previous turn. This setting seems to be a bit unnatural to me.
- The environment is constrained to Python only.


Suggestion:
- It would have been interesting to have an ablation without the compiler feedback.
- Other relevant paper to cite: https://arxiv.org/pdf/2306.09896

**Questions:**

- Could you clarify that the number of samples for the single-turn baseline (i.e., first column in Table 3 and Table 4) is the same as the number of maximum turns?
- Currently, it seems that for a given example, the same combination of feedback is given at each turn. Have you considered varying the type of feedback at different turns (e.g., compiler only in turn 1, compiler and execution in turn 2, compiler and novice feedback in turn 3, compiler and expert feedback in turn 4). Perhaps certain types of feedback or more useful (and more practical) at different steps?

---

> ### Author Response · Authors · 2024-11-22
> **Author Response to Reviewer gBAk (Q1, Q2, W1, W2, S1, S2)**
>
> We sincerely thank you for your detailed review.
> > Q1. Could you clarify that the number of samples for the single-turn baseline (i.e., first column in Table 3 and Table 4) is the same as the number of maximum turns?
>
> In our experiments, the single-turn baseline generates a single code sample via greedy decoding, following [1]. This approach mirrors the typical user experience in code generation tools, where a user receives a single initial suggestion and may refine it through feedback in subsequent turns. We will explicitly state this in the revised version.
>
>
> > Q2. Currently, it seems that for a given example, the same combination of feedback is given at each turn. Have you considered varying the type of feedback at different turns (e.g., compiler only in turn 1, compiler and execution in turn 2, compiler and novice feedback in turn 3, compiler and expert feedback in turn 4). Perhaps certain types of feedback or more useful (and more practical) at different steps?
>
> Our framework accommodates varying feedback types across turns, though we opted for consistent feedback combinations to maintain result clarity. Future work will explore a feedback recommender system to optimize feedback source selection per turn.
>
>
> > W1. While I agree that having a static benchmark like ConvCodeBench is useful, I am not convinced that it can be interpreted as a multi-turn evaluation set. Namely, my understanding is that it does not evaluate whether a model can iteratively improve its own prediction. The output from the previous turn is pre-defined, and so this seems to simply be trying to repair a program (which the model under test did not generate). If it fails to do it at a given turn, the output of that turn is ignored, and in the next turn, a fixed program is provided, which could be very different from the program the model under test generated in the previous turn. This setting seems to be a bit unnatural to me.
>
> While awaring these discrepancies, we found that appropriate reference model selection produces high correlations with live results (we provide detailed analysis in our response to 3Fw3).
>
>
> > W2. The environment is constrained to Python only.
>
> - Although ConvCodeWorld extends BigCodeBench, a benchmark focused on Python, it is not restricted to Python. Specifically:
>     - Compiler feedback can be generated using the compiler of any target language.
>     - Execution feedback is universally applicable across different programming languages.
>     - Verbal feedback remains independent of the programming language in use.
> - As ConvCodeWorld is open-sourced, it is easily adaptable for future efforts to incorporate additional programming language benchmarks.
>
>
> > S1. It would have been interesting to have an ablation without the compiler feedback.
>
> While we plan to include full ablation results in the camera-ready version, our current findings from Tables 3 and 4 show minimal performance differences between no feedback and compiler-only feedback, as most evaluated models achieve nearly 100% compilation success. This suggests that results without compiler feedback will likely remain consistent.
>
>
> > S2. Other relevant paper to cite: https://arxiv.org/pdf/2306.09896
>
> Thank you for bringing this relevant paper to our attention. We will include it in our revised manuscript.
>
>
> [1] Bei Chen, Fengji Zhang, Anh Nguyen, Daoguang Zan, Zeqi Lin, Jian-Guang Lou, & Weizhu Chen (2023). CodeT: Code Generation with Generated Tests. In The Eleventh International Conference on Learning Representations .

---

### Official Review · Reviewer_3Fw3 · 2024-10-25

**Soundness:** 3
**Presentation:** 3
**Contribution:** 2
**Rating:** 5
**Confidence:** 3

**Summary:**

This paper introduces ConvCodeWorld, a benchmark to evaluate LLMs in solving programming problems via multi-turn interactions with diverse feedback combinations. The benchmark encompasses 9 distinct feedback scenarios that combine compilation feedback, execution feedback with varying test coverage, and real-time verbal feedback simulated at different expertise levels using GPT-4o.  Additionally, a static version called ConvCodeBench is proposed, which uses pre-generated feedback logs to reduce computational costs while maintaining a high correlation with the live benchmark. In experiments, a comprehensive set of open/closed-source LLMs are evaluated on ConvCodeWorld and ConvCodeBench.

**Strengths:**

1. Evaluating LLMs in interactive environments is a significant and emerging area of study.
2. The diverse combination of different feedback is reasonable.
3. The introduction of static alternative ConvCodeBench reduces the overhead of real-time feedback generation making it scalable and practical for large-scale experiments.
4. The authors conduct comprehensive experiments covering 17 open- and closed-source LLMs.
5. The paper is easy-to-follow.

**Weaknesses:**

1. In the MINT paper (Wang et al., 2024), the multi-turn interaction (Figure 1, page 3) already includes execution results and human feedback. The human feedback in MINT covered both novice feedback (referred to as "lazy user" feedback in MINT) and expert feedback (referred to as "natural language" feedback in MINT). Given this, the novelty of this paper appears limited. Could the author include more comparisons between ConvCodeWorld and MINT to enhance clarity and novelty?
    - Reference: Wang, X., Wang, Z., Liu, J., Chen, Y., Yuan, L., Peng, H., & Ji, H. MINT: Evaluating LLMs in Multi-turn Interaction with Tools and Language Feedback. In The Twelfth International Conference on Learning Representations.

2. Discrepancy in feedback simulation compared to actual human feedback
    - Figure 13 (appendix) shows one example of the simulated expert-level verbal feedback, which is overly detailed and well-structured. I am concerned that it might not accurately reflect how real experts provide feedback when using LLMs in coding. An analysis comparing simulated feedback to actual human feedback would be both helpful and necessary.

    - A similar concern was found in the simulated novice-level verbal feedback (Figure 12, appendix). The simulated novice-level feedback includes information that seems unrealistic for novice programmers (e.g., "Also, I think there exists a simpler way to sort a list in Python."). How could a novice programmer provide such feedback based on the compilation and execution results? As suggested above, an analysis of how the simulated feedback relates to actual human feedback is necessary.

3. Lack of Clarity in Evaluation Setup: The process of using partial execution feedback was not clearly explained. Did the author randomly omit test cases from the list of tests with full coverage? If so, what criteria were applied to ensure that the selection of test cases was reasonable? Providing a detailed experimental setup would enhance clarity.

4. The claim that "Weaker Models with Feedback Surpassing Single-Turn SOTA Models" seems an unfair comparison.  While I understand that the aim is to highlight the value of multi-turn interactions, weaker models with multi-turn feedback inherently benefit from additional input context compared to single-turn, stronger models. It is therefore unsurprising that the weaker model with multi-turn interactions outperforms single-turn, stronger models. A fair comparison would evaluate the same LLMs in both single-turn and multi-turn scenarios (I thought this could also be concluded from the results section). The statement about using the same LLM would more effectively emphasize the role of multi-turn interactions in enhancing performance.

5. Bais in Static ConvCodeBench
    - The use of pre-generated feedback logs from a reference model seems biased. The errors made by the reference model might not reflect the actual errors the target model would encounter on the same problem. Thus, successful error correction might not be attributed to the feedback (or multi-turn interaction) on the code generated by the reference model, as the target model may already be capable of solving the problem on its own.
    - The correlation calculated between ConvCodeWorld and ConvCodeBench is unreasonable based on the MRR and Recall since the results may be affected by the bias mentioned above. A better way to show this correlation is to show the correlation of error patterns existing in the generated code between the reference model and the target model.

**Questions:**

1. Could the author add more comparisons between ConvCodeWorld and MINT to enhance the clarity?
2. Could the author address the discrepancy in feedback simulation compared to actual human feedback?
3. Could the author explain the evaluation setup with partial execution feedback included? A follow-up question would be what if the problem does not have full coverage tests? How many of such cases exist in the dataset?
4. Could the author respond to the potential bias in the static benchmark?  A better way to show the correlation between ConvCodeWorld and ConvCodeBench might involve presenting the error patterns between the reference model and the target model.

---

> ### Author Response · Authors · 2024-11-22
> **Author Response to Reviewer 3Fw3 (Q1, W1, Q2, W2)**
>
> We sincerely thank you for your detailed review.
> > Q1: Could the author add more comparisons between ConvCodeWorld and MINT to enhance the clarity?
>
> > W1: In the MINT paper (Wang et al., 2024), the multi-turn interaction (Figure 1, page 3) already includes execution results and human feedback. The human feedback in MINT covered both novice feedback (referred to as "lazy user" feedback in MINT) and expert feedback (referred to as "natural language" feedback in MINT). Given this, the novelty of this paper appears limited. Could the author include more comparisons between ConvCodeWorld and MINT to enhance clarity and novelty?
>
> **A1: Novelty from MINT**
> - We respectfully disagree that MINT's "lazy user" feedback serves the purpose of introducing **"engaged"** novice feedback. ConvCodeWorld's novice feedback provides verbalized explanations of other feedback (e.g. execution feedback), as illustrated in Figure 16:
>
>         "It seems like there is an issue with the socket connection or the way the code is handling the socket. The OSError exceptions are being raised during the execution of the task_func function."
>
>     In contrast, MINT only provides generic response:
>
>         "Your answer is wrong."
>
> - Another key distinction is, supporting **partial to full test coverage** in execution feedback:
>     - ConvCodeWorld allows isolated assessment of an LLM's test generalization and test utilization:
>         - **Test generalization**: The model's ability to produce code that passes comprehensive tests when only partial tests are provided.
>         - **Test utilization**: The model's capability to leverage test results for code refinement.
>     - Meanwhile, MINT (partial test only) provides an entangled evaluation of test generalization and test utilization. We revisit this point in our response to the third question (A3-2).
>
> Further detailed contrasts are summarized in Table 6 in the draft and also in response to cHoy.
>
> > Q2: Could the author address the discrepancy in feedback simulation compared to actual human feedback?
>
> > W2: Figure 13 (appendix) shows one example of the simulated expert-level verbal feedback, which is overly detailed and well-structured. I am concerned that it might not accurately reflect how real experts provide feedback when using LLMs in coding. An analysis comparing simulated feedback to actual human feedback would be both helpful and necessary.
> > A similar concern was found in the simulated novice-level verbal feedback (Figure 12, appendix). The simulated novice-level feedback includes information that seems unrealistic for novice programmers (e.g., "Also, I think there exists a simpler way to sort a list in Python."). How could a novice programmer provide such feedback based on the compilation and execution results? As suggested above, an analysis of how the simulated feedback relates to actual human feedback is necessary.
>
> **A2: Figures 12 and 13 are in-context examples, not model-generated feedback**
> - **Model-generated examples** of novice and expert feedback is presented in **Figures 16 and 17** respectively, **not in Figures 12 and 13**. Particularly, in Figure 16, the novice feedback primarily consists of verbalized explanations of the execution feedback, which we might expect from novice programmers.
>
> - We acknowledge the value of conducting a large-scale HCI study with real experts to provide deeper insights. To support such analyses in future research, we have open-sourced both ConvCodeWorld and ConvCodeBench, enabling researchers to compare simulated feedback with actual human feedback in subsequent studies.
> - Additionally, as reported from a human study in the MINT paper (Table 5), ground truth code-conditioned expert feedback simulations are generally considered  human-like, indirectly supporting the validity of  simulated feedback in large-scale evaluations.

---

> > ### Author Response · Authors · 2024-11-22
> > **Author Response to Reviewer 3Fw3 (Q3, W3, Q4)**
> >
> > > Q3-1: Could the author explain the evaluation setup with partial execution feedback included? A follow-up question would be what if the problem does not have full coverage tests? How many of such cases exist in the dataset?
> >
> > > W3: Lack of Clarity in Evaluation Setup: The process of using partial execution feedback was not clearly explained. Did the author randomly omit test cases from the list of tests with full coverage? If so, what criteria were applied to ensure that the selection of test cases was reasonable? Providing a detailed experimental setup would enhance clarity.
> >
> > > Q3-2: A follow-up question would be what if the problem does not have full coverage tests? How many of such cases exist in the dataset?
> >
> > **A3-1: Partial execution feedback setup**
> >
> > - **Selection of Test Cases**: For partial execution feedback, we consistently used the first three test cases for each problem. This approach aligns with benchmarks like HumanEval [1] and CodeContests [2], which also provide up to three public test cases.
> >
> > - **Rationale**: The first three test cases were selected to reflect a realistic scenario where general cases, often annotated early, cover main functionalities, while edge cases are typically added later. This setup simulates an incremental feedback environment, reflecting common software development workflows
> >
> > - **Controlled Selection**: Test case selection was not random. The "first three" criterion ensured a controlled and consistent evaluation across all problems, avoiding biases that could arise from arbitrary omissions.
> >
> > **A3-2. How many cases fail to provide  full tests?**
> >
> > - As mentioned in line 270 of our paper, ConvCodeWorld extends BigCodeBench, which ensures approximately **99% branch coverage** over ground truth code through its provided test cases.
> > - Even in the rare instances (fewer than 1% of cases) where full coverage is not achieved, the available test cases still provide richer execution feedback compared to partial coverage. This is particularly critical for evaluating both test generalization and test utilization, as highlighted in our response to your first question (A1).
> >
> >
> > > Q4: Could the author respond to the potential bias in the static benchmark? A better way to show the correlation between ConvCodeWorld and ConvCodeBench might involve presenting the error patterns between the reference model and the target model.
> > > W5: The use of pre-generated feedback logs from a reference model seems biased. The errors made by the reference model might not reflect the actual errors the target model would encounter on the same problem. Thus, successful error correction might not be attributed to the feedback (or multi-turn interaction) on the code generated by the reference model, as the target model may already be capable of solving the problem on its own.
> > > The correlation calculated between ConvCodeWorld and ConvCodeBench is unreasonable based on the MRR and Recall since the results may be affected by the bias mentioned above. A better way to show this correlation is to show the correlation of error patterns existing in the generated code between the reference model and the target model.
> >
> > **A4: High correlations when bias is controlled**
> > - We were aware of this bias and specifically focused on finding the right setting to address it.
> > - To reduce bias, we find the use of the weakest model as the reference model—specifically **CodeLlama-7B-Instruct**—effectove, with high correlations with the live setting, as detailed in Lines 234–251, Figure 2, and Appendix E of our paper.
> > - Our conjecture is that the weakest model generates code significantly far from the correct solution. Evaluating a model's ability to correct such code towards the correct answer provides insight into its capacity to refine its own outputs.
> > - In contrast, stronger models produce code closer to the correct solution, making the correction process less challenging and less indicative of the model's error-correcting abilities, or higher bias.
> >     - As shown in **Figure 10e**, when we use a high-performing reference model like GPT-4-0613 and provide the most informative feedback combination—$\langle f_c, f_e^*, f_v^* \rangle$—, the Spearman rank correlation decreases to **0.49**, which is significantly lower than when using CodeLlama-7B-Instruct (**0.91**; see Figure 6e) as the reference model in our setting.
> > - **Regarding the correlation of error patterns**:
> >     - While defining error patterns is challenging due to their problem-specific nature, we believe that analyzing how correlations change with different reference models effectively addresses the bias issue. In settings where we use a minimally biased reference model like CodeLlama-7B-Instruct, the MRR and Recall-based correlations remain valid and meaningful.

---

> ### Author Response · Authors · 2024-11-22
> **Author Response to Reviewer 3Fw3 (W4)**
>
> > W4-1: The claim that "Weaker Models with Feedback Surpassing Single-Turn SOTA Models" seems an unfair comparison. While I understand that the aim is to highlight the value of multi-turn interactions, weaker models with multi-turn feedback inherently benefit from additional input context compared to single-turn, stronger models. It is therefore unsurprising that the weaker model with multi-turn interactions outperforms single-turn, stronger models.
>
> **Weaker models with additional context not always outperform SOTA**
> - Prior research reported that smaller models may struggle with additional context [3].
> - Our empirical results reflect this mixed reality, showing that while some models benefit from additional context, others do not (see Table 4):
>     - Llama-3.1-8B-Instruct improves significantly from 31.4 to 51.8 Recall with feedback ($\langle f_c, f_e^*, f_v \rangle$), surpassing GPT-4o-2024-05-13's no-feedback score (50.8).
>     - DeepSeek-Coder-6.7B-Instruct, despite a higher baseline performance (35.2 vs. 31.4), only reaches 48.2 with identical feedback, **failing to surpass** GPT-4o-2024-05-13.
> - These findings highlight the challenge  of utilizing multi-turn feedbacks and justifies our evaluation to better understand this dynamic.
>
> > W4-2: A fair comparison would evaluate the same LLMs in both single-turn and multi-turn scenarios (I thought this could also be concluded from the results section). The statement about using the same LLM would more effectively emphasize the role of multi-turn interactions in enhancing performance.
>
> **Meanwhile, comparison of same LLMs in single-turn and multi-turn is also reported**
> - In our paper, **Figure 3** illustrates how the Pass@1 score evolves over multiple turns for the same model, underscoring the impact of multi-turn interactions on performance.
>
>
> [1] Chen, Mark, et al. "Evaluating large language models trained on code." arXiv preprint arXiv:2107.03374 (2021).
>
> [2] Li, Yujia, et al. "Competition-level code generation with alphacode." Science 378.6624 (2022): 1092-1097.
>
> [3] Nelson F. Liu, Kevin Lin, John Hewitt, Ashwin Paranjape, Michele Bevilacqua, Fabio Petroni, and Percy Liang. 2024. Lost in the Middle: How Language Models Use Long Contexts. Transactions of the Association for Computational Linguistics, 12:157–173.

---

> > ### Comment · Reviewer_3Fw3 · 2024-11-24
> >
> > Thank you for the responses and clarifications! Some concerns remain:
> >
> > - Regarding the novelty concern (A1, A3-2), the author highlighted test coverage and test utilization. However, these aspects pertain solely to benchmark selection rather than addressing a fundamental novelty.
> >
> > - Regarding the feedback simulation (A2), my major concern is about the design of simulating human feedback. The in-context examples seem unrealistic and could lead the LLM to produce similarly unrealistic responses, especially for expert-level feedback. Including a human evaluation would be valuable to demonstrate how feasible it is for humans to produce such fine-grained expert-level feedback to guide the LLM in error correction. In the MINT paper, a different style of simulated feedback was generated, and their human preference results can not directly or indirectly support the author's claims.
> >
> > - Regarding the bias concern in static ConvCodeBench (A4), it remains unclear how demonstrating the correlation between MRR and Recall mitigates bias in the benchmark. Specifically, when using a weaker model as the reference, a high correlation may not necessarily reflect the usefulness of the pre-generated flawed code. Instead, it could be due to the stronger model's ability to resolve the problem independently, reconstructing the solution from scratch rather than refining the existing errors. In such cases, the task essentially becomes a live benchmark, as the pre-generated code is ignored. This raises the question: why is a static version still necessary?

---

> > > ### Author Response · Authors · 2024-11-26
> > > **Author Response to Reviewer 3Fw3**
> > >
> > > > Regarding the novelty concern (A1, A3-2), the author highlighted test coverage and test utilization. However, these aspects pertain solely to benchmark selection rather than addressing a fundamental novelty.
> > >
> > > We respectfully disagree that our novelty comes from "benchmark selection" of BigCodeBench's full test annotations, but rather to enable contrasting partial and full results, enabling an evaluation of diverse aspects of model strength. Specifically:
> > > 1. LLMs that perform well on partial annotations but fail on full annotations (e.g., DeepSeek-Coder-6.7B-Instruct) reveal limitations in test utilization, an insight not observable in prior work.
> > > 2. Conversely, LLMs consistently achieving high scores (e.g., Llama-3.1-8B-Instruct) demonstrate strength in both test generalization and test utilization.
> > >
> > > These observations provide a deeper understanding of model behavior beyond what has been previously explored.
> > >
> > >
> > > > Regarding the feedback simulation (A2), my major concern is about the design of simulating human feedback. The in-context examples seem unrealistic and could lead the LLM to produce similarly unrealistic responses, especially for expert-level feedback. Including a human evaluation would be valuable to demonstrate how feasible it is for humans to produce such fine-grained expert-level feedback to guide the LLM in error correction. In the MINT paper, a different style of simulated feedback was generated, and their human preference results can not directly or indirectly support the author's claims.
> > >
> > >
> > > The reviewer expressed concerns about the realism of simulated expert-level user feedback, noting that in-context examples might lead to unrealistic responses. Following the suggestion for human evaluation, we report our findings:
> > > - Two human evaluators rated randomly assigned feedback samples from either real user feedback from [ShareGPT](https://huggingface.co/datasets/anon8231489123/ShareGPT_Vicuna_unfiltered) logs or expert feedback generated by ConvCodeWorld using GPT-4o (see Figure 18 in the updated pdf for the annotation platform). As shown in Table B, our generated feedback was found to be comparable to authentic logs in terms of expert-human-likeness and was rated higher for helpfulness, consistent with MINT's findings.
> > > - While our evaluation was limited to 20 examples (10 per source) due to time constraints of rebuttal, we plan to expand this study significantly for the camera-ready version.
> > >
> > >
> > > |Expert Feedback by|Is Helpful|Is Human-Expert-Like|
> > > |---|---|---|
> > > |ShareGPT|35%|30%|
> > > |ConvCodeWorld|55%|25%|
> > >
> > > Table B. Human evaluation of simulated expert-level user feedback by GPT-4o and real user feedback by ShareGPT.
> > >
> > >
> > >
> > > > Regarding the bias concern in static ConvCodeBench (A4), it remains unclear how demonstrating the correlation between MRR and Recall mitigates bias in the benchmark. Specifically, when using a weaker model as the reference, a high correlation may not necessarily reflect the usefulness of the pre-generated flawed code. Instead, it could be due to the stronger model's ability to resolve the problem independently, reconstructing the solution from scratch rather than refining the existing errors. In such cases, the task essentially becomes a live benchmark, as the pre-generated code is ignored. This raises the question: why is a static version still necessary?
> > >
> > > The reviewer questioned how demonstrating a correlation between MRR and Recall mitigates bias in the benchmark. We clarify that we do not propose correlation as a means of bias mitigation. Rather, we suggest it as a **proxy** indicator for identifying bias.
> > >
> > > In scenarios where live benchmarks are cost-prohibitive—for example, requiring up to $215 for verbal feedback generation as discussed in Appendix A.2—this proxy enables a performance-cost tradeoff, allowing practitioners to make informed decisions. While this serves as a proxy measure and rare corner cases exist (e.g., a weak reference model accidentally correlating with poorly generated code), this design trade-off offers a practical compromise. Additionally, it offers reproducibility and scalability in settings where live evaluations are resource-intensive or impractical.

---

### Official Review · Reviewer_cHoy · 2024-11-04

**Soundness:** 3
**Presentation:** 1
**Contribution:** 3
**Rating:** 5
**Confidence:** 3

**Summary:**

The authors introduce ConvCodeWorld, a novel framework designed to benchmark the code generation performance of large language models (LLMs) in a conversational context. This setting allows LLMs to receive various types of feedback, including compiler messages, execution outputs, and human-like verbal feedback. To support reproducibility, the authors developed a benchmark, ConvCodeBench, that uses ConvCodeWorld with pre-generated feedback logs. They conducted a comprehensive evaluation of state-of-the-art (SOTA) LLMs and a detailed ablation study, providing valuable insights to inform future research on conversational code generation.

**Strengths:**

The primary strength of this paper lies in its comprehensive approach:

- The authors cover a broad spectrum of models, including proprietary and open-weight LLMs.
- They thoroughly explore feedback settings across multiple types—compilation, execution, and verbal.
- Extensive ablation studies and analyses reveal findings that highlight key factors for advancing conversational coding tasks.

**Weaknesses:**

- **Novelty:** While this paper offers valuable insights into evaluating conversational code generation, the topic itself is not new. Prior works like InterCode and MINT have explored compilation, execution, and verbal feedback mechanisms (see Table 6).
- **Clarity of Writing:** The paper’s clarity could be improved, particularly in the introduction. Expanding the background on conversational code generation tasks and typical settings would help readers appreciate the unique contributions of this work. Additionally, certain terminology, like "partial coverage" in Table 1, could be more explicitly explained.
- **Experiment Details:** Some important experimental details are missing or unclear:
    1. **Temperature settings** for experiments are not specified. This detail is relevant as multi-turn interactions could benefit from more varied output.
    2. The **construction of the proposed dataset** is insufficiently explained, particularly given the reported 29% success rate on BigCodeBench. This contrasts with single-turn performance on ConvCodeWorld, which appears higher.
    3. **Prompting methods** lack clear definition. Both Section 4.1 and Appendix B are vague on this, with no explicit description of the prompt setup for experiments.
    4. The construction of $f_v$ and $f_v^*$ is not well explained. Examples in Section 2.1.1 and Appendix F could be enhanced with a more formal, detailed breakdown of these prompt components.
     5. In Section A.3, could the authors clarify the meaning of "a model referenced ground truth code in $f_v^*$"? Why would the model reference ground truth, and could examples illustrate this behavior?

**Questions:**

- While the authors evaluate verbal feedback with proprietary GPT models, it would be beneficial to include SOTA open-weight models as well. Given that GPT models evolve frequently, this reliance may risk data leakage. Testing open-weight models, even if currently less performant, could strengthen the reproducibility and discussion of findings.
- The claim in Appendix A.2 that the approach achieves "1.5% of the cost of human annotation" seems optimistic. Beyond the token generation cost, quality and accuracy of the model-generated content should also be factored in.

---

> ### Author Response · Authors · 2024-11-22
> **Author Response to Reviewer cHoy (Q1, Q2)**
>
> We sincerely thank you for your detailed review.
>
> > Q1. While the authors evaluate verbal feedback with proprietary GPT models, it would be beneficial to include SOTA open-weight models as well. Given that GPT models evolve frequently, this reliance may risk data leakage. Testing open-weight models, even if currently less performant, could strengthen the reproducibility and discussion of findings.
>
> **A1: Using open-weight models as verbal feedback simulators is also effective**
> - Table A supports the feasibility of using **Llama-3.1-70B-Instruct** as a **verbal feedback simulator**, replacing GPT-4o-2024-05-13.
> - At the time of setting up our experiments, we considered open-weight models. However, given MINT's findings that model performance significantly impacts feedback quality (Table 4 in the MINT paper), and the unavailability of powerful models like Llama-3.1-70B-Instruct at the time, we opted for GPT models.
>
> In the camera-ready, we will also report the main results of ConvCodeWorld and ConvCodeBench where verbal feedback is simulated by concurrent open-source models like Llama-3.1-70B-Instruct.
>
> |$v_f^*$ Generation| (Code Generation by) GPT-4o-2024-05-13 |Llama3.1-70B-Instruct|
> |--|--|--|
> | w/o Feedback | 50.8 | 45.4 |
> | GPT-4o-2024-05-13 |   64.2 | **65.1**|
> | Llama3.1-70B-Instruct  |   **65.8** | 62.1|
>
> Table A. : Pass@1 results over different expert-level verbal feedback $v_f^*$ generation on ConvCodeWorld where $\Omega = \langle f_c, \phi, f_v^* \rangle$, the total number of turns $n=1$.
>
> > Q2. The claim in Appendix A.2 that the approach achieves "1.5% of the cost of human annotation" seems optimistic. Beyond the token generation cost, quality and accuracy of the model-generated content should also be factored in.
>
> **A2: 1.5% is not optimistic**
> - We respectfully disagree 1.5% is an optimistic estimate,
> as the hourly rate of expert programmers required for $f_v^*$ annotation is indeed higher than the base rate used in the calculation.
>
> - While other costs like quality are not included in our numeric cost analysis, recent work evidences that top-performing LLMs demonstrate skills comparable to median human programmers [1], significantly reducing dependence on human experts. Furthermore, MINT’s human evaluation (Table 5) confirms that expert feedback simulations conditioned on ground truth code are capable of producing feedback that is both helpful and human-like, underscoring the feasibility of automating this process.

---

> ### Author Response · Authors · 2024-11-22
> **Author Response to Reviewer cHoy (W1, W2-1)**
>
> > W1: **Novelty:** While this paper offers valuable insights into evaluating conversational code generation, the topic itself is not new. Prior works like InterCode and MINT have explored compilation, execution, and verbal feedback mechanisms (see Table 6).
>
> We elaborate distinctive implications from InterCode and MINT:
> 1. Comparative Analyses of **Partial to Full Test Coverage** in Execution Feedback enables to evaluate both:
>
>     - **Test generalization**: A model's ability to produce code that passes full tests even when only partial tests are provided.
>     - **Test utilization**: A model's capability to leverage given test results for code refinement.
>
>     a) MINT (partial test only): an **entangled evaluation** of test generalization and test utilization.
>
>     b) InterCode (full test only): evaluates **test utilization only**.
>
>     c) **ConvCodeWorld, by providing both partial and full test**, enables isolated evaluation of each test generalization and test utilization as we illustrate below.
>
>     For instance, in Table 4:
>     - DeepSeek-Coder-6.7B-Instruct:
>         -  Modest test generalization ($\langle f_c, \phi, \phi \rangle \rightarrow \langle f_c, f_e, \phi \rangle$: 35.2 $\rightarrow$ 37.7)
>         -  But limited test utilization ($\langle f_c, f_e, \phi \rangle \rightarrow \langle f_c, f_e^*, \phi \rangle$: 37.7 $\rightarrow$ 37.5)
>     -  In contrast, Qwen1.5-72B-Chat exhibits strong capabilities in both aspects:
>         - Test generalization: $\langle f_c, \phi, \phi \rangle \rightarrow \langle f_c, f_e, \phi \rangle$: 33.2 $\rightarrow$ 39.9
>         - Test utilization: $\langle f_c, f_e, \phi \rangle \rightarrow \langle f_c, f_e^, \phi \rangle$: 39.9 $\rightarrow$ 47.5
>
> 2. ConvCodeWorld simulates  an **"engaged"** user, offering verbalized explanations of test results, as illustrated in Figure 16:
>
>
>         "It seems like there is an issue with the socket connection or the way the code is handling the socket. The `OSError` exceptions are being raised during the execution of the `task_func` function."
>
>    In contrast, InterCode lacks verbal feedback, and MINT provides only generic feedback:
>
>         "Your answer is wrong."
>
>     **Full execution feedback + novice feedback** scenario in ConvCodeWorld effectively evaluates **how verbalized explanations enhance models' test utilization capabilities**. In Table 4:
>     - Full test coverage execution feedback ($\langle f_c, f_e^*, \phi \rangle$): Llama-3.1-8B-Instruct's test utilization capabilities (40.0) are weaker compared to CodeQwen1.5-7B-Chat (**41.1**).
>     -  However, the inclusion of novice feedback ($\langle f_c, f_e^*, f_v \rangle$): significantly improves Llama-3.1-8B-Instruct's performance, surpassing CodeQwen1.5-7B-Chat (**51.8** vs. 49.5).
>
> 3. Covering comprehensive combinations of feedback types,
>  ConvCodeWorld analyzes previously underexplored cases, such as:
>     - Full execution feedback vs. partial execution feedback + novice feedback
>     - Partial execution feedback + expert feedback vs. full execution feedback + expert feedback
>     - Full execution feedback + novice feedback vs. expert feedback
>
> 4. Cost-Effective Static Benchmark (ConvCodeBench): ConvCodeBench correlates strongly with online evaluation while reducing costs. Neither MINT nor InterCode provide such a static benchmark.
>
> > W2-1: **Clarity of Writing:** The paper’s clarity could be improved, particularly in the introduction. Expanding the background on conversational code generation tasks and typical settings would help readers appreciate the unique contributions of this work.
>
> We will expand the introduction with comprehensive background on conversational code generation, and relocate related work to follow it—better highlighting our novel contributions.

---

> > ### Author Response · Authors · 2024-11-22
> > **Author Response to Reviewer cHoy (W2-2, W3)**
> >
> > > W2-2: Additionally, certain terminology, like "partial coverage" in Table 1, could be more explicitly explained.
> >
> > We will clarify the term "partial coverage" and other terminology in Table 1's caption.
> >
> > > W3: **Experiment Details:** Some important experimental details are missing or unclear:
> >     1. **Temperature settings** for experiments are not specified. This detail is relevant as multi-turn interactions could benefit from more varied output.
> >     2. The **construction of the proposed dataset** is insufficiently explained, particularly given the reported 29% success rate on BigCodeBench. This contrasts with single-turn performance on ConvCodeWorld, which appears higher.
> >     3. **Prompting methods** lack clear definition. Both Section 4.1 and Appendix B are vague on this, with no explicit description of the prompt setup for experiments.
> >     4. The construction of f_v and f_v^* is not well explained. Examples in Section 2.1.1 and Appendix F could be enhanced with a more formal, detailed breakdown of these prompt components.
> >     5. In Section A.3, could the authors clarify the meaning of "a model referenced ground truth code in f_v^∗"? Why would the model reference ground truth, and could examples illustrate this behavior?
> >
> > 1. **Temperature settings**: We used greedy decoding (temperature = 0) in all experiments, following [2]. We will include this detail in the revised manuscript.
> > 2. **Dataset construction**: The reported Pass@1 of 29% on BigCodeBench refers to its hard subset. Our experiments are based on the **full set**, where success rates are higher (see the [BigCodeBench leaderboard](https://huggingface.co/spaces/bigcode/bigcodebench-leaderboard) and select "full set"). This distinction and additional dataset details will be clarified.
> > 3. **Prompting methods**:  Although we provided a link to our [runnable codebase](https://anonymous.4open.science/r/convcodeworld-B6A8/README.md) on our [anonymous homepage](https://huggingface.co/spaces/ConvCodeWorld/ConvCodeWorld), Sections 2.1.1 and Appendix F will be enhanced with detailed breakdowns and examples.
> > 4. **Construction of** $f_v$ **and** $f_v^*$: We will enhance Sections 2.1.1 and Appendix F with a more formal and detailed breakdown of these prompt components, including illustrative examples.
> > 5. **Meaning of *"a model referenced ground truth code"***: refers to cases when the  feedback simulator mentions ground truth code provided in the prompt, illustrates in examples:
> >
> >     - Example 1 (desirable): Feedback guides toward the ground truth without explicitly referencing it.
> >     - Example 2 (undesirable): Feedback directly references ground truth, leading to "leakage"—Details of how we control leakage can be found in Appendix A.2 in the draft and also in response to Reviewer acNZ.
> >
> >
> > Example 1.
> > ```
> > 1. **Configuration File Reading**: The `previous_code` correctly reads the configuration file using `configparser`. However, ensure that the configuration file path is valid and exists before attempting to read it. This is not explicitly checked in the `previous_code`.
> >
> > 2. **Directory Existence Check**: The `previous_code` uses `os.path.exists(project_dir)` to check if the project directory exists. While this works, it is more appropriate to use `os.path.isdir(project_dir)` to specifically check for directory existence, as it is more semantically correct.
> >
> > 3. **ZIP Archive Creation**: The `previous_code` attempts to create the ZIP archive using `shutil.make_archive(project_dir, 'zip', archive_dir)`. This is incorrect because `shutil.make_archive` expects the base name of the archive and the root directory to archive. The correct usage should be `shutil.make_archive(base_name=os.path.splitext(zip_file_path)[0], format='zip', root_dir=project_dir)`.
> >
> > 4. **Exception Handling**: The `previous_code` raises a generic `Exception` if the ZIP archive creation fails. While this is acceptable, it is better to provide a more specific error message indicating the failure reason. Additionally, ensure that the ZIP file is actually created by checking its existence after the `shutil.make_archive` call.
> >
> > 5. **Return Value**: The `previous_code` correctly returns `True` if the ZIP archive is successfully created. However, it should also ensure that the ZIP file exists before returning `True`.
> >
> > 6. **Code Simplicity and Readability**: The `previous_code` includes a detailed docstring, which is good practice. However, the actual implementation can be simplified and made more readable by following the correct usage of `shutil.make_archive` and ensuring proper exception handling.
> >
> > Overall, the `previous_code` has the right structure but needs corrections in the directory existence check, ZIP archive creation, and exception handling to function correctly.
> > ```

---

> > > ### Author Response · Authors · 2024-11-22
> > > **Author Response to Reviewer cHoy (W3)**
> > >
> > > Example 2.
> > > ```
> > > 1. **Class Name**: The class name in the `previous_code` is `EmailHandler`, but it should be `EmailRequestHandler` to match the `ground_truth_code`.
> > >
> > > 2. **Content-Type Check**: Instead of directly checking the `Content-Type` header, use `cgi.parse_header` to parse the header and then check if `ctype != 'application/json'`.
> > >
> > > 3. **Error Handling for Content-Type**: When the `Content-Type` is not `application/json`, simply send a 400 response and end headers without writing a message to the response body.
> > >
> > > 4. **Reading Content-Length**: Use `length = int(self.headers.get('content-length'))` instead of `content_length = int(self.headers.get('Content-Length', 0))`.
> > >
> > > 5. **JSON Decoding**: When catching `json.JSONDecodeError`, send a 400 response and end headers without writing a message to the response body.
> > >
> > > 6. **Missing Fields Check**: When required fields are missing, send a 400 response and end headers without writing a message to the response body.
> > >
> > > 7. **SMTP Authentication Error Handling**: When catching `smtplib.SMTPAuthenticationError`, send a 535 response and end headers without writing a message to the response body.
> > >
> > > 8. **General Exception Handling**: Remove the general exception handler that sends a 500 response, as it is not present in the `ground_truth_code`.
> > >
> > > By making these changes, the `previous_code` will align more closely with the `ground_truth_code`.
> > > ```
> > >
> > >
> > > [1] Li, Yujia, et al. "Competition-level code generation with alphacode." Science 378.6624 (2022): 1092-1097.
> > >
> > > [2] Bei Chen, Fengji Zhang, Anh Nguyen, Daoguang Zan, Zeqi Lin, Jian-Guang Lou, & Weizhu Chen (2023). CodeT: Code Generation with Generated Tests. In The Eleventh International Conference on Learning Representations.

---

> > > > ### Comment · Reviewer_cHoy · 2024-11-23
> > > >
> > > > Thank you to the authors for providing thorough clarifications. I especially appreciate the inclusion of experiments using open-weight models as verbal feedback simulators, which enhances the reproducibility and robustness of the benchmark.
> > > > I also commend the authors for addressing concerns regarding writing clarity, experimental details, and terminology. With these improvements, the paper is more polished and easier to follow.
> > > >
> > > > That said, some concerns remain. For instance, while ConvCodeWorld refines certain implementation issues observed in previous works like InterCode and MINT, the fundamental differences between these approaches remain unclear. However, I do appreciate the authors’ thorough ablations and their discussion of findings, which provide valuable insights into this domain.  Additionally, the claim that using LLMs for labeling achieves "1.5% of the cost of human annotation" still feels insufficiently rigorous.
> > > >
> > > > With these improvements and the responses provided, I would love to raise my rating to 5.

---

> > > > > ### Author Response · Authors · 2024-11-26
> > > > > **Author Response to Reviewer cHoy**
> > > > >
> > > > > Thank you for your thoughtful comment and for acknowledging how our thorough ablations and analysis provide valuable insights into this domain.
> > > > >
> > > > > Please note that we have further clarified the distinction of ConvCodeWorld to previous works in our recent comment towards Reviewer 3Fw3.

---

### Meta-Review · Area_Chair_a3th · 2024-12-16

**Metareview:**

This paper introduces ConvCodeWorld, a benchmark designed to assess the ability of large language models (LLMs) to solve programming problems through multi-turn interactions featuring diverse feedback mechanisms. The benchmark includes nine distinct feedback scenarios, combining compilation feedback, execution feedback with varying test coverage, and simulated real-time verbal feedback modeled at different expertise levels using GPT-4. To complement this, the authors propose a static counterpart, ConvCodeBench, which employs pre-generated feedback logs to minimize computational overhead while maintaining a strong correlation with the live benchmark.

The study evaluates a broad spectrum of open- and closed-source LLMs on both ConvCodeWorld and ConvCodeBench, offering comprehensive experimental results. While conversational approaches to code generation are widely acknowledged, benchmarks in this area remain relatively underdeveloped. This work contributes a valuable resource for the community to explore this crucial problem, delivering meaningful insights through its thorough and diverse evaluations.

**Additional Comments On Reviewer Discussion:**

NA

---

### Decision · Program_Chairs · 2025-01-22

Accept (Poster)